# VecMol: Vector-Field Representations for 3D Molecule Generation

Yuchen Hua [* 1]   Xingang Peng [* 1 2]   Jianzhu Ma [3 4]   Muhan Zhang [1 5]

## Abstract

Generative modeling of three-dimensional (3D) molecules is a fundamental yet challenging problem in drug discovery and materials science. Existing approaches typically represent molecules as 3D graphs and co-generate discrete atom types with continuous atomic coordinates, which requires maintaining mutual consistency between the inferred bond graph and the sampled coordinates under geometric perturbations—a tight coupling between heterogeneous modalities. We propose VecMol, a paradigm-shifting framework that reimagines molecular representation by modeling 3D molecules as continuous vector fields over Euclidean space, where vectors point toward nearby atoms and implicitly encode molecular structure. The vector field is parameterized by a neural field and generated using a latent diffusion model, avoiding explicit graph generation and decoupling structure learning from discrete atom instantiation. Experiments on the QM9 and GEOM-Drugs benchmarks validate the feasibility of this novel approach, suggesting vector-field-based representations as a promising new direction for 3D molecular generation.

## 1. Introduction

Generative modeling of three-dimensional (3D) molecular structures has emerged as an important problem in drug discovery and materials science. Unlike domains with canonical grid-based representations, molecular systems admit a wide range of structural representations, ranging from discrete graphs to continuous spatial descriptions. As a result, the choice of representation plays a central role in the design of molecular generative models. While recent advances in diffusion models and equivariant architectures have demonstrated promising performance on molecular generation tasks (Hoogeboom et al., 2022b; Geiger & Smidt, 2022), existing approaches continue to face a trade-off between structural fidelity and computational tractability.

A prevailing paradigm represents molecules as point clouds of atoms processed by equivariant Graph Neural Networks (GNNs) (Satorras et al., 2022). Such models naturally capture local chemical environments and symmetries, yet their expressivity is constrained by the locality of message passing (Xu et al., 2019; Morris et al., 2019), and their computational cost typically scales quadratically with the number of atoms. Moreover, point-cloud-based generative models often require an explicit upper bound on molecular size, introducing artificial cardinality constraints during training and sampling. In contrast, voxel-based approaches discretize 3D space into regular grids (Pinheiro et al., 2024), enabling the use of highly expressive architectures such as Transformers with global attention. However, these methods suffer from severe memory and computational overheads, which scale cubically with the molecular size and spatial resolution, rendering them impractical for high-resolution molecular modeling.

In this work, we propose a new representation that models molecules as *continuous vector fields* defined over three-dimensional space. Instead of treating a molecule as a set of atoms, our approach represents it as a neural field that maps any spatial location to a vector pointing toward nearby atomic centers. This formulation is well aligned with the physical nature of molecular systems, whose interactions and energy landscapes are inherently continuous functions of space. By parameterizing these vector fields with a conditional neural field architecture, we decouple global structural encoding from local geometric realization: a compact latent code captures the overall molecular structure, while a shared $E(n)$-equivariant decoder maps spatial coordinates to field values. Building upon this representation, we introduce VECMOL, a two-stage generative framework. In the first stage, a neural field autoencoder compresses molecular structures into a fixed-dimensional latent space. In the second stage, a Latent Diffusion Probabilistic Model (LDPM) is trained to model the distribution of these latent codes,

---

[*]Equal contribution [1]Institute for Artificial Intelligence, Peking University [2]School of Intelligence Science and Technology, Peking University [3]Department of Electronic Engineering, Tsinghua University [4]Institute for AI Industry Research, Tsinghua University [5]State Key Laboratory of General Artificial Intelligence, Peking University. Correspondence to: Muhan Zhang <muhan@pku.edu.cn>.

*Proceedings of the 43rd International Conference on Machine Learning*, Seoul, South Korea. PMLR 306, 2026. Copyright 2026 by the author(s).

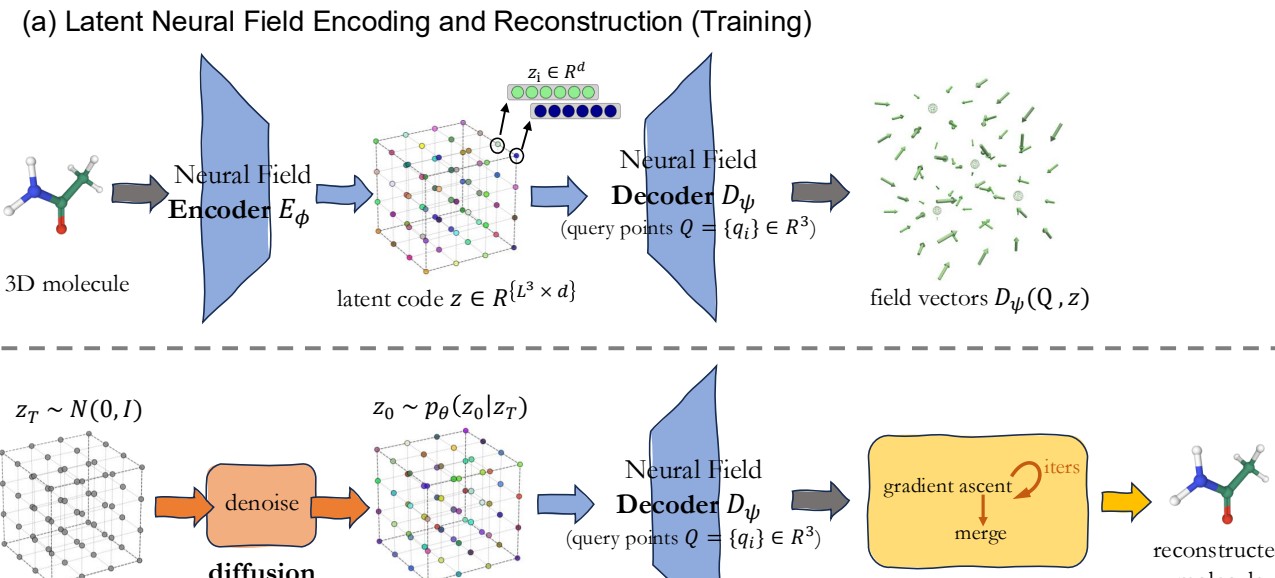

(a) Latent Neural Field Encoding and Reconstruction (Training)

(b) Latent Field Diffusion and Molecular Generation (Sampling)

*Figure 1.* **Overview of the proposed neural field framework for 3D molecular modeling.** The figure illustrates two tightly coupled pipelines that share the same neural field decoder and reconstruction module. **Top: Latent neural field encoding and reconstruction.** A 3D molecule, represented by its atomic coordinates and types, is first encoded by a neural field encoder $E_\phi$ into a grid-based latent field $\mathbf{z} \in \mathbb{R}^{L^3 \times d}$, where each spatial location stores a local latent code. Given a set of spatial query points $Q = \{\mathbf{q}_i\}_{i=1}^m \in \mathbb{R}^{m \times 3}$, a neural field decoder $D_\psi$ maps the latent field to a continuous molecular vector field $\mathbf{V} = D_\psi(Q, \mathbf{z})$. **Bottom: Latent field diffusion and molecular generation.** A denoising diffusion probabilistic model is trained in the latent field space. Starting from Gaussian noise $\mathbf{z}_T \sim \mathcal{N}(\mathbf{0}, \mathbf{I})$, the diffusion model progressively denoises the latent variables to obtain a sampled latent field $\mathbf{z}_0$. Th neural field decoder and the reconstruction module then converts the sampled vector field into a discrete molecular structure through iterative gradient-based ascent and merging operations (see Section 3.4).

enabling the generation of novel latent codes, which are decoded into vector fields of novel molecules. Importantly, the dimensionality of the latent space is **independent of the number of atoms**. After the vector field is decoded, atomic positions emerge implicitly through solving an ordinary differential equation (ODE) problem on random initial points. This design avoids predefined limits on molecular size and offers a principled alternative to diffusion over raw atomic coordinates.

Our main contributions are summarized as follows:

- We introduce a continuous vector-field representation of 3D molecules that overcomes the discretization limitations of point-cloud and voxel-based representations.
- We propose VECMOL, a two-stage generative framework that combines neural field autoencoding with latent diffusion, decoupling molecular generation from explicit atomic cardinality constraints.
- We demonstrate that molecular structures can be generated via diffusion on a compact, structure-aware latent space, followed by optimization-based decoding from an implicit neural field.
- We present a systematic analysis of the new generation

pipeline, including its generation behavior, the robustness of the the neural field, and the formulation of vector field.

Our code is publicly available at `https://github.com/MuLabPKU/VecMol`. Beyond static molecule generation, the continuous vector-field representation also opens the door to modeling dynamics such as molecular dynamics simulations and structure prediction, where the learned field can serve as a prior for conformational exploration and refinement.

## 2. Related Work

**Equivariant 3D molecular generation.** A substantial body of work represents molecules as point clouds or 3D graphs, where atoms are modeled as nodes with continuous spatial coordinates and discrete chemical types, and learns generative models that respect Euclidean symmetries (Gasteiger et al., 2022). Representative approaches include autoregressive coordinate generation (Gebauer et al., 2020), equivariant normalizing flows (Köhler et al., 2020), and, more recently, diffusion-based models. Among these, E(3)-equivariant diffusion has emerged as a dominant paradigm. EDM (Hoogeboom et al., 2022a), for example,

applies equivariant diffusion to atomic point clouds and is widely adopted as a baseline for unconditional 3D molecular generation (Peng et al., 2023; Vignac et al., 2023; Hua et al., 2024). Despite their strong empirical performance, coordinate-based methods must jointly generate discrete atom types and continuous coordinates, rely on expressive yet computationally intensive equivariant GNNs (Schütt et al., 2021) , and typically assume a predefined number of atoms. In contrast, our approach models molecular structure using a continuous vector field defined over space, avoiding explicit coordinate-wise generation and naturally decoupling molecular geometry from atom indexing and ordering.

**Molecular generation with volumetric and field-based representations.** An alternative class of methods represents molecules as spatial fields rather than explicit sets of atomic coordinates (Schütt et al., 2018). Voxel-based approaches discretize 3D space into regular grids and encode atoms as smooth density functions, enabling convolutional architectures for learning and generation (Gebauer et al., 2020). Early work in structure-based drug design (Ragoza et al., 2020) and more recent models such as VoxMol (Pinheiro et al., 2024) show that denoising voxelized molecular densities can yield competitive performance in 3D molecule generation. However, voxel-based representations remain discretized in space, and accurate generation often requires high spatial resolution, leading to substantial computational and memory costs. Field-based methods alleviate this limitation by modeling molecules as continuous atomic density fields. For instance, FuncMol (Kirchmeyer et al., 2025) applies score-based generative modeling to continuous density representations and recovers all-atom structures via walk–jump sampling. While these approaches primarily model scalar-valued density fields, our method instead represents molecules using vector-valued gradient fields, which encode directional information toward atomic centers and provide a richer structural signal.

**Neural-field representations and generative modeling.** Neural fields (also known as implicit neural representations) model signals (such as 3D shapes, images, or physical fields) as continuous functions over spatial coordinates parameterized by neural networks (Sitzmann et al., 2020), and have become a standard tool for high-resolution 3D geometry. Seminal works such as DeepSDF (Park et al., 2019) and Occupancy Networks (Mescheder et al., 2019) represent shapes using scalar signed distance or occupancy functions, while NeRF (Mildenhall et al., 2020) extends this paradigm to view-dependent scene modeling. A key property is that neural fields can be conditioned on a latent code and decoded into continuous spatial signals, enabling scalable learning across large datasets with a shared decoder (Dupont et al., 2022b). Building on this foundation, recent work explores generative modeling over neural fields, either by applying diffusion directly in function space (Dupont et al., 2022a; Babu et al., 2025) or by performing generative modeling in learned latent spaces (Chen & Zhang, 2019; Kim et al., 2023; Zhang et al., 2023). Our work adopts this neural-field generation paradigm for molecular modeling, but departs from prior approaches by learning probabilistic models over vector-valued neural fields, enabling the explicit modeling of directional information and increased representational expressiveness.

## 3. Method

Figure 1 illustrates an overview of the proposed framework. We represent molecules as continuous vector fields defined over three-dimensional space and model their distribution using a two-stage generative process. Given an input molecular structure, we first encode it into a compact latent representation via a neural field autoencoder. Conditioned on this latent code, the decoder reconstructs a continuous vector field, from which atomic positions can be recovered at arbitrary spatial resolution. To enable molecular generation, we train a denoising diffusion probabilistic model (DDPM) in the latent space and sample novel latent codes, which are then decoded into molecular vector fields and subsequently converted into discrete atomic structures through a refinement step.

### 3.1. The Vector-Field Representation

We represent a 3D molecule as a continuous vector field defined over 3D space, which encodes atomic occupancy and geometric structure. Unlike discrete representations such as point clouds or voxel grids, this formulation models molecular geometry as a continuous function, enabling resolution-free reconstruction and scalable modeling of molecules with varying sizes and atom counts. We emphasize the distinction between the latent representation and the field it parameterizes: the latent tensor $\mathbf{z} \in \mathbb{R}^{L^3 \times d}$ lives on a finite grid and is itself discrete, but the decoder $D_\psi$ defines a continuous function over $\mathbb{R}^3$ that can be evaluated at arbitrary continuous query locations $\mathbf{q}$, allowing the model to capture geometric structures at any resolution.

Formally, we define a vector field as a mapping $\mathbf{v} : \mathbb{R}^3 \to \mathbb{R}^{K \times 3}$, where each spatial query point $\mathbf{q} \in \mathbb{R}^3$ is mapped to $K$ three-dimensional vectors, one for each atom element type. Each vector $\mathbf{v}_k(\mathbf{q})$ points along a direction leading to the nearest atom of type $k$ (i.e., following an ascent path toward its location), providing a continuous indication of atomic locations.

Given a 3D molecule, let $\mathcal{A}_k = \{\mathbf{a}_j\}_{j=1}^{n_k}$ denote the set of $n_k$ atoms of type $k$ with coordinates $\mathbf{a}_j \in \mathbb{R}^3$. For a query point $\mathbf{q} \in \mathbb{R}^3$ and atom type $k$, we compute distances $d_j = \|\mathbf{q} - \mathbf{a}_j\|$ and direction vectors $\mathbf{d}_j = \mathbf{a}_j - \mathbf{q}$. We

further define a distance clip $d_{\text{clip}}$ and $\tilde{d}_j = \min(d_j, d_{\text{clip}})$. The ground-truth vector field for atom type $k$ at query point $\mathbf{q}$ is then defined as:

$$\mathbf{v}_k^*(\mathbf{q}) = \sum_{j=1}^{n_k} w_j^{\text{softmax}} \cdot w_j^{\text{mag}} \cdot \hat{\mathbf{d}}_j. \tag{1}$$

where

$$w_j^{\text{softmax}} = \frac{\exp(-d_j/\sigma_{\text{sf}})}{\sum_{j'=1}^{n_k} \exp(-d_{j'}/\sigma_{\text{sf}})},$$

$$w_j^{\text{mag}} = \exp\left(-\frac{\tilde{d}_j^2}{2\sigma_{\text{mag}}^2}\right) \cdot \tilde{d}_j, \tag{2}$$

$$\hat{\mathbf{d}}_j = \frac{\mathbf{d}_j}{\|\mathbf{d}_j\| + \epsilon}.$$

Here, $\hat{\mathbf{d}}_j$ is a normalized direction vector pointing from $\mathbf{q}$ toward atom $\mathbf{a}_j$, where $\epsilon > 0$ ensures numerical stability. The magnitude term $w_j^{\text{mag}}$ controls the contribution strength of each atom and yields bounded, well-conditioned vectors by suppressing distant interactions through a clipped Gaussian function, parameterized by $\sigma_{\text{mag}}$ and $d_{\text{clip}}$. The softmax weight $w_j^{\text{softmax}}$ selects dominant directions by emphasizing nearby atoms while maintaining smooth transitions across space, where the temperature $\sigma_{\text{sf}}$ controls the locality of this selection. See Appendix B.3 for a detailed discussion of the field design objectives. For absent atom types, we define a complementary repulsive field $\mathbf{v}_k^*(\mathbf{q}) = w_{\text{rep}}(\mathbf{q}) \cdot (\mathbf{q} - \mathbf{c})/\|\mathbf{q} - \mathbf{c}\|$ that generates outward-pointing vectors away from the molecular centroid $\mathbf{c}$, where $w_{\text{rep}}(\mathbf{q}) = \gamma \exp(-\|\mathbf{q} - \mathbf{c}\|^2/2\sigma_{\text{rep}}^2)$ (see Appendix B.3).

Unless otherwise specified, we use $\sigma_{\text{sf}} = 0.1$, $\sigma_{\text{mag}} = 0.45$, and $d_{\text{clip}} = 0.8$ for all experiments. A detailed analysis of field variants and hyperparameter sensitivity is provided in Appendix B.3, Appendix C.2 and Appendix C.2. Our field formulation explicitly decouples direction selection from magnitude control, which is crucial for stable optimization and accurate structure recovery. This representation yields a continuous, resolution-independent, and atom-count-agnostic molecular representation.

### 3.2. Neural Field Autoencoder

Given the ground-truth vector-field representations described in Section 3.1, we train a neural field autoencoder that maps molecular structures to compact latent representations and reconstructs continuous vector fields from them. The autoencoder consists of (i) an encoder that embeds variable-size molecular structures into fixed-size latent field codes defined on a regular grid, and (ii) a decoder that reconstructs continuous vector fields from latent codes and spatial query points.

### 3.2.1. ENCODER: MOLECULE-TO-FIELD ENCODING

The encoder $E_\phi$ maps a 3D molecule to a latent field representation. Given the atomic coordinates $\mathbf{X} \in \mathbb{R}^{n \times 3}$ and atom types $T \in \{0, \ldots, K-1\}^n$ for a molecule with $n$ atoms, the encoder produces latent codes $\mathbf{z} \in \mathbb{R}^{L^3 \times d}$ defined on a regular 3D grid of $L^3$ anchor points.

To bridge the variable-size molecular graph and the fixed grid representation, we adopt a cross-graph encoding architecture with two types of edges: (i) an intra-atomic graph $\mathcal{G}_a$ connecting each atom to its $k_a$ nearest atomic neighbors via k-NN, and (ii) a cross graph $\mathcal{G}_{ag}$ connecting each grid anchor point to its $k_g$ nearest atoms. A cross-graph message-passing architecture aggregates atomic features into grid-aligned latent codes based on spatial proximity, preserving molecular geometry while enabling scalable, fixed-dimensional encoding. Architectural details are provided in Appendix A.1.

### 3.2.2. DECODER: FIELD PREDICTION WITH EQUIVARIANCE

The decoder reconstructs a continuous, atom-type-specific vector field from latent field codes by querying arbitrary spatial locations. Formally, given a set of query points $Q = \{\mathbf{q}_i\}_{i=1}^m$, where $\mathbf{q}_i \in \mathbb{R}^3$, and a latent grid representation $\mathbf{z} \in \mathbb{R}^{L^3 \times d}$, the decoder outputs a vector field:

$$\mathbf{V} = D_\psi(Q, \mathbf{z}) \in \mathbb{R}^{m \times K \times 3}, \tag{3}$$

where $\mathbf{V}[i, k, :]$ denotes the predicted vector at query point $\mathbf{q}_i$ for atom type $k$.

To ensure physical consistency, the decoder is parameterized by an $\text{E}(n)$-equivariant graph neural network (EGNN), which guarantees equivariance to 3D rotations and invariance to global translations. Query points and latent grid anchors are treated as nodes in a local interaction graph, allowing the decoder to interpolate latent field information in a geometry-aware manner. Architectural details of the EGNN are provided in Appendix A.1.

Instead of directly regressing vector values, the decoder predicts a virtual source location $\mathbf{s}_i^{(k)} \in \mathbb{R}^3$ for each query point $\mathbf{q}_i$ and atom type $k$. The vector field is then defined as the displacement from the query point to the predicted source:

$$\mathbf{v}_k(\mathbf{q}_i) = \mathbf{s}_i^{(k)} - \mathbf{q}_i. \tag{4}$$

This formulation has two advantages: (1) It naturally enforces translation equivariance, as the field depends only on relative positions. (2) It stabilizes learning by constraining vectors to point toward implicit attractors, which aligns with the physical interpretation of atomic centers as sinks of the field.

After the final EGNN layer, the virtual source is obtained by applying an atom-type-specific equivariant prediction head to the query node representation. The complete procedure preserves $E(n)$-equivariance by construction, ensuring that the decoded vector field transforms consistently under rigid motions of the molecule.

### 3.2.3. TRAINING

We train the encoder $E_\phi$ and decoder $D_\psi$ to reconstruct ground-truth vector fields from molecular structures. Given a molecule with atomic coordinates $\mathbf{X}$ and atom types $T$, we sample $m$ spatial query points $Q = \{\mathbf{q}_i\}_{i=1}^m$ within a bounding region, encode the molecule to obtain latent field codes $\mathbf{z} = E_\phi(\mathbf{X}, T)$, and predict vector-field values $\mathbf{V} = D_\psi(Q, \mathbf{z})$ at the query locations.

The training objective minimizes the mean squared error between predicted and ground-truth fields:

$$\mathcal{L}_{\text{auto}} = \frac{1}{mK} \sum_{i=1}^m \sum_{k=1}^K \|\mathbf{v}_k(\mathbf{q}_i) - \mathbf{v}_k^*(\mathbf{q}_i)\|^2, \qquad (5)$$

where $\mathbf{v}_k^*(\mathbf{q}_i)$ is the ground-truth field defined in Section 3.1.

### 3.3. Latent Diffusion Model

We model the distribution of molecular structures by performing generative modeling in the latent space learned by the neural field autoencoder. Specifically, we train a denoising diffusion probabilistic model (DDPM) over latent codes $\mathbf{z}_0 \in \mathbb{R}^{L^3 \times d}$, whose dimensionality is fixed across molecules.

The forward diffusion process gradually corrupts latent codes with Gaussian noise over $T$ timesteps:

$$q(\mathbf{z}_t|\mathbf{z}_0) = \mathcal{N}(\mathbf{z}_t; \sqrt{\bar{\alpha}_t}\mathbf{z}_0, (1 - \bar{\alpha}_t)\mathbf{I}), \qquad (6)$$

with $\bar{\alpha}_t$ following a cosine schedule (see Appendix A.2 for details). The reverse denoising process is modeled as

$$p_\theta(\mathbf{z}_{t-1}|\mathbf{z}_t) = \mathcal{N}(\mathbf{z}_{t-1}; \boldsymbol{\mu}_\theta(\mathbf{z}_t, t), \sigma_t^2\mathbf{I}), \qquad (7)$$

where the parameterization of $\boldsymbol{\mu}_\theta$ and $\sigma_t^2$, as well as additional architectural details of $\epsilon_\theta$, are provided in Appendix A.2.

The denoiser $\epsilon_\theta$ is implemented as an equivariant GNN (EGNN) over grid anchor points. Node features are initialized from latent codes $\mathbf{z}_t$, and timestep embeddings are injected to condition the network on the diffusion step $t$. Edges are constructed using radius-based neighborhoods.

By performing diffusion in the latent space of continuous molecular vector fields, our approach avoids directly modeling the complex correspondence between continuous Euclidean geometry and discrete atomic types. Instead, the diffusion model operates on a structured latent representation that faithfully encodes the underlying vector-field space.

After training the autoencoder, the encoder $E_\phi$ is frozen. Latent codes $\mathbf{z}_0 = E_\phi(\mathbf{X}, T)$ extracted from training molecules are used to train the diffusion model, with the standard noise prediction objective:

$$\mathcal{L}_{\text{diff}} = \mathbb{E}_{t, \mathbf{z}_0, \epsilon} \left[ \|\epsilon - \epsilon_\theta(\sqrt{\bar{\alpha}_t}\mathbf{z}_0 + \sqrt{1 - \bar{\alpha}_t}\epsilon, t)\|^2 \right], \quad (8)$$

where $\epsilon \sim \mathcal{N}(\mathbf{0}, \mathbf{I})$ and $t$ is uniformly sampled from $\{1, 2, \ldots, T\}$.

### 3.4. Molecule Sampling and Reconstruction

Given a trained decoder that produces a continuous vector field $\mathbf{V} = D_\psi(Q, \mathbf{z})$, we reconstruct discrete molecular structures by iteratively extracting atomic candidates from the field. The reconstruction procedure corresponds to the final module in Figure 1. While higher-order ODE solvers are applicable, we find that a simple Euler integration scheme is sufficient in practice.

To generate molecules, we first sample a latent code via the reverse diffusion process starting from Gaussian noise $\mathbf{z}_T \sim \mathcal{N}(\mathbf{0}, \mathbf{I})$. The reverse diffusion process iteratively denoises the latent code:

$$\mathbf{z}_{t-1} = \frac{1}{\sqrt{\alpha_t}} \left( \mathbf{z}_t - \frac{\beta_t}{\sqrt{1 - \bar{\alpha}_t}} \epsilon_\theta(\mathbf{z}_t, t) \right) + \sigma_t \epsilon, \quad (9)$$

for $t = T, T - 1, \ldots, 1$, where $\epsilon \sim \mathcal{N}(\mathbf{0}, \mathbf{I})$ for $t > 1$ and $\epsilon = \mathbf{0}$ for $t = 1$.

The denoised latent code $\mathbf{z}_0$ is then decoded into a continuous vector field $\mathbf{V} = D_\psi(Q, \mathbf{z}_0)$ for query points $Q$. Discrete atomic positions are recovered by initializing candidate query points and performing gradient ascent along the predicted vector field for each atom type $k$:

$$\mathbf{q}_i^{(t+1)} = \mathbf{q}_i^{(t)} + \eta \cdot \mathbf{v}_k(\mathbf{q}_i^{(t)}), \qquad (10)$$

where $\eta > 0$ is the step size. The iteration stops either when the field norm falls below a threshold, $\|\mathbf{v}_k(\mathbf{q}_i^{(t)})\| < \tau$, or when a maximum number of iterations $T_{\max}$ is reached, to ensure convergence and avoid potential infinite loop.

The converged points are clustered using DBSCAN with distance threshold $\epsilon$ and minimum sample size $m_{\min}$ to obtain final atomic coordinates. Chemical bonds are inferred using standard cheminformatics software (OpenBabel (O'Boyle et al., 2011)) and our post-processing in Appendix A.3. A detailed step-by-step description of the reconstruction procedure is provided in Appendix A.3 and further illustrated from an intuitive and algorithmic perspective in Appendix B.1.

# 4. Experiments

We now evaluate our model for unconditional generation. We start with a description of our experimental setup (Section 4.1), then present our results on three datasets (Section 4.2).

## 4.1. Experimental setup

**Datasets.** We evaluate VECMOL on three datasets: *QM9* (Wu et al., 2018), *GEOM-drugs* (Axelrod & Gómez-Bombarelli, 2022), and *CREMP* (Kirchmeyer et al., 2025) (cyclic peptides). We model hydrogen explicitly and consider five chemical elements for QM9 (C, H, O, N, F) and eight for GEOM-drugs (C, H, O, N, F, S, Cl and Br). For CREMP, we follow the same evaluation protocol as Func-Mol, considering six atom types (C, H, O, N, F, S). We follow the standard data splits and preprocessing protocols established in the previous work (Vignac et al., 2023).

**Implementation details.** VECMOL follows the auto-encoding framework described in Section 3. Latent codes are produced by an 8-layer GNN encoder that maps molecular graphs onto a regular 3D grid of anchor points with spacing 3.0 Å. For **QM9**, we use grid size $L = 5$ with MLP hidden width 1024; for **GEOM-drugs**, we use grid size $L = 7$ with MLP hidden width 1024. The latent code dimension $d = 384$ is shared across all datasets. For **CREMP**, we use grid size $L = 8$ with MLP hidden width 384, and the decoder is a 6-layer EGNN-based neural field. EGNN is used to ensure $E(n)$-equivariance in vector field prediction. Models are trained end-to-end with random rotational augmentation.

For molecule generation, we train a latent diffusion model on the learned vector field manifold. The denoiser is a 10-layer EGNN with hidden dimension 512, using radius-based graph construction with radius set to $1.8\times$ the anchor spacing. We adopt a cosine noise schedule with $T = 1000$ steps and an $\mathbf{x}_0$-prediction objective for stable training.

**Baselines.** We compare VECMOL to four state-of-the-art approaches. *EDM* (Hoogeboom et al., 2022b) and *GeoLDM* (Xu et al., 2023) are diffusion models operating on point clouds (the latter is a latent-space extension of the former). *VoxMol* (Pinheiro et al., 2024) is a voxel-based generative model that uses neural empirical Bayes, similar to our generative approach. *FuncMol* (Schneuing et al., 2022) is a field-based method that represents molecules as neural fields. All of the methods generate molecules as a set of atom types and their coordinates. EDM and GeoLDM apply diffusion directly to point clouds, while VoxMol, FuncMol and VECMOL rely on an additional (cheap) post-processing step to extract atomic coordinates from voxel grids or modulation codes, respectively. We follow previous work (Ragoza

et al., 2020; Vignac et al., 2023; Pinheiro et al., 2024; Guan et al., 2023; Schneuing et al., 2022), and use standard chem-informatics software (OpenBabel (O'Boyle et al., 2011)) to determine the molecule's atomic bonds based on atomic coordinates and our post-processing Section A.3.

**Metrics.** Following prior work (Pinheiro et al., 2024), we evaluate unconditional molecule generation on QM9 and GEOM-drugs using: *stable mol* and *stable atom*, the fractions of stable molecules and atoms (Hoogeboom et al., 2022b); *validity*, the fraction of molecules passing RD-Kit (Landrum, 2016) sanitization; *uniqueness*, the proportion of valid molecules with distinct canonical SMILES; *valency $W_1$*, the Wasserstein distance between generated and test valency distributions; *atoms TV* and *bonds TV*, the total variation distances of atom and bond type distributions; *bond length $W_1$* and *bond angle $W_1$*, the Wasserstein distances of bond length and angle distributions.

For GEOM-drugs, we further assess conformational quality and molecular properties using: *single fragment*, the fraction of single-fragment molecules; *median strain energy* (Harris et al., 2023), computed as the energy gap between the generated and UFF-relaxed conformations using RDKit (Rappé et al., 1992); *ring size TV* and *number of atoms/mol TV*, measuring distributional discrepancies in ring sizes and molecule sizes (largest fragment only); and *QED, SA and logp*, evaluating drug-likeness (Bickerton et al., 2012), synthesizability (Ertl & Schuffenhauer, 2009), and lipophilicity (via RDKit).

**Ablation studies.** We conduct extensive ablation studies to validate the design choices in VECMOL. Appendix C.1 analyzes the impact of autoencoder capacity and grid resolution, showing that reconstruction fidelity is the primary performance bottleneck. Appendix B.2 compares vector-field against scalar-valued representations, demonstrating that directional information significantly improves reconstruction accuracy. Appendix B.3 ablates candidate field definitions and validates the Gaussian-Clip formulation. Appendix C.3 and C.4 examine sensitivity to reconstruction parameters and the diffusion neighborhood radius, respectively.

## 4.2. Generation Results

We evaluate the generative performance of VECMOL on QM9 and GEOM-drugs and compare it against representative baselines. In addition to the default generation setting, where molecules are generated by sampling latent codes from noise, we report two auxiliary reconstruction variants: $\text{VECMOL}_{NF}$ uses latent codes obtained by encoding a ground-truth molecule through the autoencoder encoder, then decodes them back to a vector field for reconstruction. This variant involves no diffusion sampling and thus reflects the autoencoder's reconstruction quality, serving as an up-

*Table 1.* QM9 generation results w.r.t. the test set for 10,000 samples per model. ↑/↓ indicate that higher/lower is better. The row *data* corresponds to randomly sampled molecules from the validation set.

| | stable mol %$_\uparrow$ | stable atom %$_\uparrow$ | valid %$_\uparrow$ | unique %$_\uparrow$ | valency $W_{1\downarrow}$ | atom TV$_\downarrow$ | bond TV$_\downarrow$ | bond len $W_{1\downarrow}$ | bond ang $W_{1\downarrow}$ | time (s)$_\downarrow$ |
|---|---|---|---|---|---|---|---|---|---|---|
| *data* | 98.7 | 99.8 | 98.9 | 99.9 | .001 | .003 | .000 | .000 | .120 | – |
| EDM | 97.9 | 99.8 | 99.0 | 98.5 | .011 | .021 | .002 | .001 | .440 | 6.93 |
| GeoLDM | 97.5 | 99.9 | 100. | 98.0 | .005 | .017 | .003 | .007 | .435 | 9.66 |
| VoxMol | 89.3 | 99.2 | 98.7 | 92.1 | .023 | .029 | .009 | .003 | 1.96 | 14.18 |
| FuncMol | 89.2 | 99.0 | 100. | 92.8 | .021 | .012 | .006 | .005 | 1.56 | – |
| VECMOL | 97.5 (±0.1) | 99.8 (±0.0) | 98.4 (±0.2) | 98.1 (±0.2) | 0.038 (±0.003) | 0.011 (±0.001) | 0.006 (±0.000) | 0.024 (±0.003) | 1.01 (±0.03) | 14.94 |
| VECMOL$_{NF}$ | 97.6 | 99.8 | 98.6 | 99.9 | .023 | .009 | .004 | .019 | 0.90 | – |
| VECMOL$_{Diff}$ | 97.4 | 99.8 | 98.4 | 99.7 | .037 | .011 | .006 | .022 | 0.99 | – |

*Table 2.* GEOM-drugs generation results (standard metrics) w.r.t. the test set for 10,000 samples per model. ↑/↓ indicate that higher/lower is better. The row *data* corresponds to randomly sampled molecules from the validation set.

| | stable mol %$_\uparrow$ | stable atom %$_\uparrow$ | valid %$_\uparrow$ | unique %$_\uparrow$ | valency $W_{1\downarrow}$ | atom TV$_\downarrow$ | bond TV$_\downarrow$ | bond len $W_{1\downarrow}$ | bond ang $W_{1\downarrow}$ | time (s)$_\downarrow$ |
|---|---|---|---|---|---|---|---|---|---|---|
| *data* | 99.9 | 99.9 | 99.8 | 100.0 | .001 | .001 | .025 | .000 | .05 | – |
| EDM | 40.3 | 97.8 | 87.8 | 99.9 | .285 | .212 | .048 | .002 | 6.42 | 79.18 |
| GeoLDM | 57.9 | 98.7 | 100. | 100. | .197 | .099 | .024 | .009 | 2.96 | 73.10 |
| VoxMol | 75.0 | 98.1 | 93.4 | 99.6 | .254 | .033 | .024 | .002 | 0.64 | 84.32 |
| FuncMol | 69.7 | 98.8 | 100 | 95.3 | .245 | .109 | .052 | .003 | 2.49 | – |
| VECMOL | 80.2 (±0.3) | 99.1 (±0.1) | 89.2 (±0.4) | 91.2 (±0.3) | 0.108 (±0.003) | 0.034 (±0.000) | 0.112 (±0.001) | 0.024 (±0.001) | 4.81 (±0.20) | 31.62 |

per bound for the full generation pipeline. VECMOL$_{Diff}$, which reconstructs molecules from vector fields decoded from diffusion-denoised latent codes. For VECMOL$_{Diff}$, we do not perform the full reverse diffusion process from $T$ to 0. Instead, we randomly sample a diffusion timestep $t$ and **apply a single denoising step** to the corresponding noisy latent code, and decode the resulting denoised code into a vector field. This setting is used to probe the quality of the learned diffusion model independently of long-horizon sampling effects. All results are reported for 10,000 generated molecules per model using the standard evaluation protocol described in Section 4.1. We report uncertainty bars (±1$\sigma$) estimated over 3 independent sampling runs.

**Results on QM9.** Table 1 reports generation results on QM9. VECMOL achieves performance comparable to strong baselines across all reported metrics, including stability, validity, and distributional distances, indicating that the proposed vector-field-based representation is sufficient to model both molecular geometry and chemical statistics for small molecules.

**Results on GEOM-Drugs.** Table 2 reports generation results on GEOM-drugs. We further evaluate generated molecules using additional indicators that capture fragmentation, strain energy, and chemical realism, summarized in

Table 3. On GEOM-drugs, we consider two autoencoder capacities: MLP hidden width 384 and 1024 (the latter used in our main results). A complete comparison of the two configurations is provided in Appendix C.1. The larger model substantially improves generation quality: validity rises from 82.5% to 89.2%, bond-related Wasserstein distances drop significantly (e.g., bond angle $W_1$ from 8.98 to 4.81), and drug-relevant metrics such as QED and SA also improve. This demonstrates that reconstruction fidelity, rather than the latent representation or diffusion process, is the primary bottleneck, and that scaling model capacity is an effective path to improving generation quality on complex molecular distributions.

**Sampling time.** Sampling times (reported in Tables 1 and 2) are measured on a single NVIDIA RTX 4090 GPU with 10,000 generated molecules per method; additional timing details and stage-wise breakdowns are provided in Appendix D.2.

**Results on CREMP.** To further validate the scalability of our vector-field representation to larger molecular systems, we evaluate VECMOL on the CREMP dataset (cyclic peptides with up to 100+ atoms, six element types: C, H, O, N, F, S). We follow the evaluation protocol of FuncMol (Kirchmeyer et al., 2025): the largest connected component is

*Table 3.* GEOM-drugs generation results (additional metrics) w.r.t. the test set for 10,000 samples per model. ↑/↓ indicate that higher/lower is better. The row *data* corresponds to randomly sampled molecules from the validation set.

| | single frag %↑ | median energy↓ | ring sz TV↓ | atms/mol TV↓ | QED ↑ | SA ↑ | logP ↑ |
|---|---|---|---|---|---|---|---|
| *data* | 100. | 54.5 | .011 | .000 | .658 | .832 | 2.95 |
| EDM | 42.2 | 951.3 | .976 | .604 | .472 | .514 | 1.11 |
| GeoLDM | 51.6 | 461.5 | .644 | .469 | .497 | .593 | 1.05 |
| VoxMol | 82.6 | 69.2 | .264 | .636 | .659 | .762 | 2.73 |
| FuncMol | 70.5 | 109.7 | .427 | 1.05 | .713 | .811 | 3.09 |
| VECMOL | 81.82 (±0.95) | 87.3 (±3.4) | 0.296 (±0.037) | 0.214 (±0.003) | 0.653 (±0.001) | 2.08 (±0.19) | 2.30 (±0.17) |

*Table 4.* Unconditional generation results on the CREMP dataset (quality and local geometry). Backbone KL divergence is reported in Table 5.

| | valid %↑ | unique %↑ | stable mol %↑ | stable atom %↑ | bond len $W_1$↓ | bond ang $W_1$↓ |
|---|---|---|---|---|---|---|
| VECMOL | 83.1 | 98.0 | 82.2 | 99.6 | 0.067 | 4.43 |

*Table 5.* Backbone KL divergence (test ∥ sampled) on the CREMP dataset (lower is better).

| | $\theta_1$ KL↓ | $\theta_2$ KL↓ | $\theta_3$ KL↓ | $\phi$ KL↓ | $\psi$ KL↓ | $\omega$ KL↓ |
|---|---|---|---|---|---|---|
| FuncMol | 0.0954 | 0.1219 | 0.1986 | 0.1997 | 0.1218 | 0.1244 |
| VECMOL | 0.0726 | 0.0887 | 0.1445 | 0.1516 | 0.0734 | 0.0971 |

retained, hydrogens are included for all metrics, and backbone KL divergence is computed using their angular binning. As shown in Table 4 and Table 5, VECMOL achieves 83.1% RDKit validity and 98.0% uniqueness, with strong geometric fidelity across backbone angles and dihedrals. Compared to FuncMol, VECMOL consistently achieves lower (better) backbone KL divergence across all six angle and dihedral types, indicating more accurate capture of the complex 3D folding geometries of cyclic peptides.

Furthermore, CREMP yields better local geometry metrics than GEOM-drugs (e.g., bond length $W_1$ of 0.067 vs. 0.024, bond angle $W_1$ of 4.43 vs. 4.81), likely due to its fewer atom types and highly regular repeating backbone motifs, which present a favorable setting for our type-wise continuous field representation. Despite the smaller autoencoder capacity, VECMOL achieves strong results on CREMP, likely because the dataset consists of macrocyclic peptides with only 6 atom types and highly regular repeating backbone motifs, which are easier to model from limited data.

### 4.3. Case Analysis

We provide a case analysis to intuitively illustrate the spatial structure of the gradient vector field and its effect on

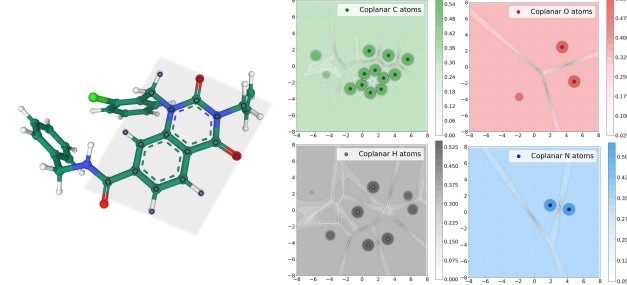

*Figure 2.* Element-specific gradient magnitude on a planar cross-section of a representative molecule. High-gradient regions surround atomic nuclei, with element-dependent spatial extent and intensity.

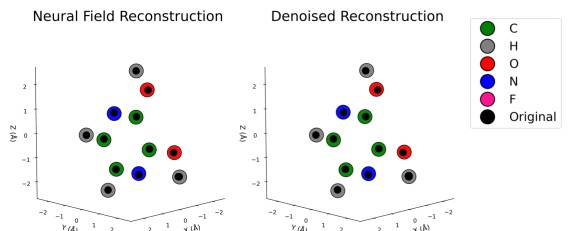

*Figure 3.* Atomic coordinate reconstruction from neural vector fields. Left: reconstruction from raw neural field codes; right: reconstruction after latent denoising.

molecular reconstruction.

**Element-specific structure of the vector field.** Figure 2 shows the gradient magnitude on a 2D planar slice of a molecule, visualized separately for different atomic elements. Only atoms within the planar are displayed. The vector field shows high-gradient regions around atoms and has zero magnitude at the atomic center, providing clear cues for query points to converge to atomic centers.

**Qualitative reconstruction accuracy.** Figures 3 and 4 present reconstruction results for the same molecule at the coordinate and geometric levels. Atoms are recovered by clustering query points evolved under two versions of predicted vector field.

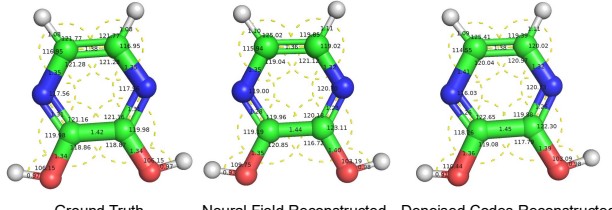

*Figure 4.* Comparison of molecular geometry. From left to right: ground truth, reconstruction from raw codes, and reconstruction from denoised codes.

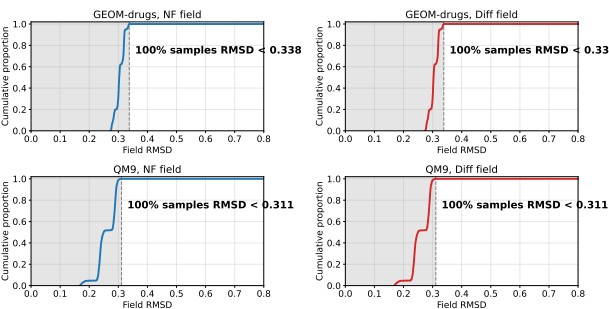

*Figure 5.* Cumulative distribution of field RMSD versus ground truth for NF and Diff on 1,000 molecules from QM9 and GEOM-drugs. Both fields closely match the ground truth.

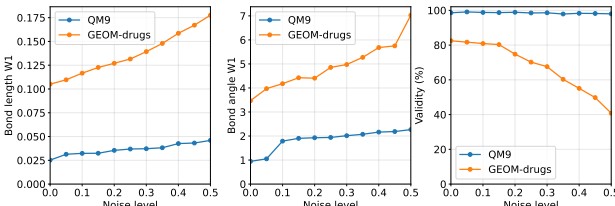

*Figure 6.* Robustness of the learned neural field to isotropic Gaussian noise in the latent code space.

Reconstruction from raw neural field codes captures the global molecular geometry, with atomic centers forming distinct spatial clusters. After latent denoising, these clusters become more compact and better aligned with the ground-truth positions. This improvement is also reflected in molecular geometry: bond lengths and bond angles in the denoised reconstruction show reduced variance and closer agreement with reference values, indicating improved structural consistency.

### 4.4. Neural Field and Diffusion Model Quality

**Quantitative Field Quality Evaluation**   We evaluate two types of predicted vector fields: NF and Diff, decoded from neural and diffusion-denoised latent codes, respectively. Figure 5 shows the cumulative distribution of RMSD (root-mean-square deviation) between predicted and ground-truth fields on 1,000 molecules from QM9 and GEOM-drugs, demonstrating close agreement.

**Neural Field Robustness**   To assess robustness, we inject isotropic Gaussian noise $\sigma_{\text{noise}} \in [0, 0.5]$ into latent codes and decode molecules from the perturbed representations. Figure 6 reports how molecular validity, bond-length, and bond-angle distribution distances change with increasing noise. The neural field remains resilient, with metrics degrading gradually, indicating that the latent representation is smooth and stable, which benefits diffusion model training.

## 5. Conclusion

We have presented VECMOL, a novel approach for unconditional 3D molecule generation that represents molecules as continuous vector-valued neural fields. By encoding directional gradient information rather than scalar densities, our method enables direct gradient-based reconstruction while maintaining geometric equivariance through E($n$)-equivariant graph neural networks. The proposed Gaussian-Clip field formulation addresses key challenges in field-to-molecule conversion, providing bounded, well-conditioned gradients that facilitate stable learning and accurate reconstruction. Our experiments demonstrate that diffusion in the latent vector-field space produces chemically valid and stable molecules, achieving competitive performance on QM9 and GEOM-drugs datasets, with particular advantages in molecular stability and geometric consistency on larger drug-like molecules.

Despite these strengths, limitations remain. Future work could explore more sophisticated field definitions, conditional generation for structure-based drug design, and extensions to larger and more complex molecular systems. Beyond de novo generation, the continuous vector-field representation also holds promise for modeling molecular dynamics and serving as a learned prior for structure prediction, where the field can guide conformational exploration and refinement. Overall, the vector-field representation offers a principled, scalable framework bridging continuous fields and discrete molecular structures, opening new avenues for molecular modeling.

## Impact Statement

This work contributes methodological advances in unconditional three-dimensional molecular generation, which is a foundational problem in molecular modeling and design. By enabling scalable modeling of larger molecular systems and more efficient sampling, the proposed approach may support downstream research in areas such as drug discovery, biology, and materials science. While practical adoption requires extensive experimental validation beyond the scope of this work, progress in this direction has the potential to facilitate scientific discovery and improve human well-being. As with all emerging machine learning technologies, careful and responsible use is essential to ensure that such developments lead to positive societal outcomes.

## Acknowledgements

This work is supported by National Natural Science Foundation of China (62550138, 62276003).

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

# A. Methodological and Architectural Details

## A.1. Autoencoder Architecture

This appendix provides the detailed architecture of our autoencoder, which maps between discrete molecular structures and continuous vector fields. The model is designed to preserve geometric consistency, ensure translation invariance, and support stable learning of 3D molecular representations.

**Encoder: From Molecular Graphs to Latent Grids.** The encoder transforms a discrete molecular structure into a structured latent representation suitable for continuous field decoding. Given a molecule with atomic coordinates

$$\mathcal{A} = \{\mathbf{a}_j \in \mathbb{R}^3\}_{j=1}^N$$

and corresponding atom types, we embed the molecule into a regular 3D grid of resolution $L \times L \times L$, where each grid cell $g$ is associated with a learnable latent vector $\mathbf{z}_g \in \mathbb{R}^d$. The full latent representation is denoted as

$$\mathbf{z} \in \mathbb{R}^{L^3 \times d}.$$

To construct grid features, atoms are connected to nearby grid anchors based on Euclidean distance, forming a bipartite graph. An E($n$)-equivariant graph neural network (EGNN) aggregates atomic information into the grid nodes, ensuring rotational equivariance and translational invariance. Distance-based cutoff and Gaussian smearing are used to smooth messages over neighboring atoms, preventing discontinuities in the latent code due to small perturbations in atomic positions. This produces a latent grid that compactly encodes both geometry and atom-type information.

**Decoder: Equivariant Vector Field Prediction.** The decoder reconstructs a continuous vector field from the latent grid. Given a set of query points

$$Q = \{\mathbf{q}_i\}_{i=1}^m, \quad \mathbf{q}_i \in \mathbb{R}^3,$$

and the latent grid $\mathbf{z}$, the decoder outputs atom-type-specific vectors

$$\mathbf{V} = D_\psi(Q, \mathbf{z}) \in \mathbb{R}^{m \times K \times 3}.$$

**Graph Construction and Node Initialization.** A bipartite graph is built between query points and their $k$ nearest grid anchors. Node features are initialized as

$$\mathbf{h}_i^{(0)} = \begin{cases} \mathbf{0}, & \text{if node } i \text{ is a query point}, \\ \mathbf{z}_g, & \text{if node } i \text{ is a grid anchor}, \end{cases}$$

and each node is associated with coordinates $\mathbf{x}_i$ corresponding to its spatial location.

**Equivariant Message Passing.** The decoder applies $L_d$ EGNN layers to propagate information across the graph. At layer $\ell$, edge messages are computed as

$$\mathbf{m}_{ij}^{(\ell)} = \text{EdgeMLP}\left([\mathbf{h}_i^{(\ell-1)}, \mathbf{h}_j^{(\ell-1)}, \|\mathbf{x}_i^{(\ell-1)} - \mathbf{x}_j^{(\ell-1)}\|]\right),$$

where $j \in \mathcal{N}(i)$ denotes neighbors of node $i$.

Coordinates are updated equivariantly:

$$\mathbf{x}_i^{(\ell)} = \mathbf{x}_i^{(\ell-1)} + \frac{1}{|\mathcal{N}(i)|} \sum_{j \in \mathcal{N}(i)} a_{ij}^{(\ell)} \frac{\mathbf{x}_j^{(\ell-1)} - \mathbf{x}_i^{(\ell-1)}}{\|\mathbf{x}_j^{(\ell-1)} - \mathbf{x}_i^{(\ell-1)}\| + \epsilon},$$

with

$$a_{ij}^{(\ell)} = \text{CoordMLP}(\mathbf{m}_{ij}^{(\ell)}).$$

Node features are updated via

$$\mathbf{h}_i^{(\ell)} = \mathbf{h}_i^{(\ell-1)} + \text{NodeMLP}\left(\mathbf{h}_i^{(\ell-1)}, \frac{1}{|\mathcal{N}(i)|} \sum_{j \in \mathcal{N}(i)} \mathbf{m}_{ij}^{(\ell)}\right).$$

**Virtual Source Prediction and Field Output.**  After the final layer, the decoder predicts a virtual source location for each query point and atom type:

$$\Delta\mathbf{x}_i^{(k)} = \frac{1}{|\mathcal{N}(i)|} \sum_{j\in\mathcal{N}(i)} a_{ij}^{(k)} \frac{\mathbf{x}_j^{(L_d)} - \mathbf{x}_i^{(L_d)}}{\|\mathbf{x}_j^{(L_d)} - \mathbf{x}_i^{(L_d)}\| + \epsilon}, \quad \mathbf{s}_i^{(k)} = \mathbf{x}_i^{(L_d)} + \Delta\mathbf{x}_i^{(k)}.$$

The decoded vector field is then defined as

$$\mathbf{v}_k(\mathbf{q}_i) = \mathbf{s}_i^{(k)} - \mathbf{q}_i,$$

which guarantees translation invariance and preserves $\mathrm{E}(n)$-equivariance, ensuring consistent field predictions under rigid motions.

## A.2. Latent Diffusion Model

**Prediction Target and Diffusion Schedule.**  We adopt an $x_0$-prediction objective in latent space, as it has been found to yield more stable training and higher-quality molecular generations compared to noise prediction. This benefit can be attributed to the bounded and smooth nature of the learned latent manifold.

The forward diffusion process gradually corrupts latent codes $\mathbf{z}_0$ over $T$ timesteps with Gaussian noise:

$$q(\mathbf{z}_t|\mathbf{z}_0) = \mathcal{N}(\mathbf{z}_t; \sqrt{\bar{\alpha}_t}\mathbf{z}_0, (1 - \bar{\alpha}_t)\mathbf{I}), \tag{11}$$

where $\bar{\alpha}_t = \prod_{s=1}^{t} \alpha_s$, $\alpha_t = 1 - \beta_t$, and $\beta_t$ follows a cosine schedule (Nichol & Dhariwal, 2021):

$$\bar{\alpha}_t = \frac{\cos((t/T + s)/(1 + s) \cdot \pi/2)^2}{\cos(s/(1 + s) \cdot \pi/2)^2}, \quad s = 0.008. \tag{12}$$

The reverse denoising process is parameterized as:

$$p_\theta(\mathbf{z}_{t-1}|\mathbf{z}_t) = \mathcal{N}(\mathbf{z}_{t-1}; \boldsymbol{\mu}_\theta(\mathbf{z}_t, t), \sigma_t^2\mathbf{I}), \tag{13}$$

with

$$\boldsymbol{\mu}_\theta(\mathbf{z}_t, t) = \frac{1}{\sqrt{\alpha_t}}\left(\mathbf{z}_t - \frac{\beta_t}{\sqrt{1 - \bar{\alpha}_t}}\epsilon_\theta(\mathbf{z}_t, t)\right), \quad \sigma_t^2 = \frac{1 - \bar{\alpha}_{t-1}}{1 - \bar{\alpha}_t}\beta_t. \tag{14}$$

Here, $\epsilon_\theta$ is the learned denoiser network, predicting either the added noise or the clean latent code (in our case $x_0$).

The diffusion model is trained to minimize the prediction error:

$$\mathcal{L}_{\text{diff}} = \mathbb{E}_{t,\mathbf{z}_0,\boldsymbol{\epsilon}}\left[\|\boldsymbol{\epsilon} - \epsilon_\theta(\sqrt{\bar{\alpha}_t}\mathbf{z}_0 + \sqrt{1 - \bar{\alpha}_t}\boldsymbol{\epsilon}, t)\|^2\right], \tag{15}$$

where $\boldsymbol{\epsilon} \sim \mathcal{N}(\mathbf{0}, \mathbf{I})$ and $t \sim \text{Uniform}\{1, 2, \ldots, T\}$.

**EGNN Denoiser Architecture.**  The denoiser $\epsilon_\theta$ is implemented as an *Equivariant Graph Neural Network (EGNN)* operating on latent grid anchor points. Each node $i$ corresponds to a grid anchor with feature initialized as

$$\mathbf{h}_i^{(0)} = \mathbf{z}_t[i, :]. \tag{16}$$

Edges are constructed using radius-based neighborhoods: two nodes $i, j$ are connected if $\|\mathbf{x}_i - \mathbf{x}_j\| < r$, where $r$ is a hyperparameter controlling local connectivity. Messages at layer $\ell$ are computed as:

$$\mathbf{m}_{ij}^{(\ell)} = \text{EdgeMLP}\left([\mathbf{h}_i^{(\ell-1)}, \mathbf{h}_j^{(\ell-1)}, \|\mathbf{x}_i - \mathbf{x}_j\|, \mathbf{t}_{\text{emb}}]\right), \tag{17}$$

and node features are updated via

$$\mathbf{h}_i^{(\ell)} = \mathbf{h}_i^{(\ell-1)} + \sum_{j\in\mathcal{N}(i)} \mathbf{m}_{ij}^{(\ell)}. \tag{18}$$

Here, $\mathbf{t}_{\text{emb}}$ is a sinusoidal embedding of the diffusion timestep $t$. This architecture ensures SE(3) equivariance while allowing flexible local message passing in the latent field.

*Table 6.* Reconstruction parameters for query-point-based field-to-molecule conversion on different datasets.

|                              | QM9  | GEOM-Drugs |
|------------------------------|------|------------|
| Iterations $n_{\text{iter}}$ | 500  | 500        |
| Step size $\eta$             | 0.1  | 0.1        |
| Clustering radius $\varepsilon$ | 0.1  | 0.1     |
| Minimum samples $N_{\text{min}}$ | 3 | 3         |
| Query points (C)             | 200  | 1000       |
| Query points (H)             | 200  | 1000       |
| Query points (O)             | 30   | 100        |
| Query points (N)             | 30   | 150        |
| Query points (F)             | 15   | 20         |
| Query points (S)             | –    | 20         |
| Query points (Cl)            | –    | 20         |
| Query points (Br)            | –    | 20         |

## A.3. Algorithmic View of Field–Molecule Conversion

**Query-Point-Based Atom Reconstruction.**  To recover discrete atomic structures from a predicted vector field, we employ a particle-based reconstruction procedure consisting of gradient-guided dynamics followed by density-based clustering.

**Gradient-Guided Query Point Dynamics.**  We initialize a large set of particles uniformly in three-dimensional space. Each particle iteratively updates its position by following the local gradient direction:

$$\mathbf{x}_{t+1} = \mathbf{x}_t + \eta\, \mathbf{g}(\mathbf{x}_t), \tag{19}$$

where $\mathbf{g}(\cdot)$ denotes the predicted gradient vector field and $\eta$ is a fixed step size. As particles move, they continuously query the vector field at updated locations, resulting in adaptive trajectories that naturally follow the geometry of the implicit potential landscape. After sufficient iterations, particles converge to equilibrium regions where the gradient magnitude becomes negligible, analogous to gradient descent toward local minima of an unknown potential.

**Density-Based Atom Extraction.**  After particle trajectories converge, their final positions form dense point clouds around atomic centers. We extract discrete atoms by applying DBSCAN with clustering radius $\varepsilon$ and minimum number of points $N_{\text{min}}$, and take the centroid of each valid cluster as the predicted atomic coordinate.

Unlike conventional density-based clustering scenarios, the quality of reconstruction in our setting is largely determined by the accuracy of particle convergence rather than the specific choice of clustering hyperparameters. Once particles have sufficiently converged to atomic equilibria, their spatial distributions become highly compact, making the resulting clustering outcomes insensitive to moderate variations in $\varepsilon$.

Complete parameter settings for each dataset are summarized in Table 6.

**Post-processing: Bond Inference and Chemical Refinement.**  The field-to-molecule reconstruction procedure yields discrete atomic coordinates and element types, but does not explicitly encode chemical bond connectivity. In addition, small geometric deviations may arise from particle-based field following and clustering, which can affect chemical validity if bonds are inferred using strict geometric criteria. To address these issues, we apply a lightweight post-processing step that combines distance-based heuristics with standard chemical rules, implemented with the assistance of Open Babel.

Candidate covalent bonds are inferred based on inter-atomic distances: two atoms are allowed to form a bond if their Euclidean distance does not exceed $\rho$ times the corresponding standard bond length, where $\rho$ is a tunable tolerance parameter. In all experiments, we set $\rho = 1.5$. This relaxed criterion improves robustness to residual coordinate noise without altering atomic identities.

For atoms with unsatisfied valence, we further examine local geometric configurations, including bond angles and coordination patterns, to estimate feasible hybridization states. If a chemically more stable configuration can be achieved by adding hydrogen atoms without violating basic valence or geometric constraints, hydrogens are automatically supplemented. This

---

**Algorithm 1** Query-Point–Based Field-to-Molecule Reconstruction

---

**Require:** Latent code $\mathbf{z}$, field decoder $D_\psi$, step size $\eta$, convergence threshold $\tau$, DBSCAN parameters $(\varepsilon, N_{\min})$, number of iterations $T$
**Ensure:** Atomic coordinates $\mathbf{X} \in \mathbb{R}^{n \times 3}$ and atom types $\boldsymbol{\tau} \in \{0, \ldots, K-1\}^n$
1: Initialize empty set of converged query points $\mathcal{Q} \leftarrow \emptyset$
2: **for** each element type $k = 0, \ldots, K-1$ **do**
3:     Initialize $n_k$ query points $\{\mathbf{q}_i^{(0)}\}_{i=1}^{n_k}$ uniformly within the bounding volume
4:     **for** $t = 1$ to $T$ **do**
5:         **for** each query point $\mathbf{q}_i^{(t-1)}$ not yet converged **do**
6:             Query field: $\mathbf{g}_k \leftarrow D_\psi(\mathbf{q}_i^{(t-1)}, \mathbf{z}, k)$
7:             **if** $\|\mathbf{g}_k\| < \tau$ **then**
8:                 Mark $\mathbf{q}_i$ as converged
9:             **else**
10:                 $\mathbf{q}_i^{(t)} \leftarrow \mathbf{q}_i^{(t-1)} + \eta \cdot \mathbf{g}_k$
11:             **end if**
12:         **end for**
13:     **end for**
14:     Add all converged query points of type $k$ to $\mathcal{Q}$
15: **end for**
16: Apply DBSCAN clustering on $\mathcal{Q}$ with radius $\varepsilon$ and minimum points $N_{\min}$
17: Extract cluster centroids as atomic coordinates $\mathbf{X}$
18: Assign atom types $\boldsymbol{\tau}$ based on element-specific query point membership
19: **return** $(\mathbf{X}, \boldsymbol{\tau})$

---

post-processing stage serves solely as chemical refinement and does not modify the learned vector-field representation or the generative model.

**Algorithmic Summary.** Algorithm 1 summarizes the complete procedure for converting learned vector fields into discrete molecular structures. The reconstruction procedure converts a continuous vector field into discrete atomic coordinates through iterative gradient ascent followed by clustering. For each atom type $k$, we initialize candidate points uniformly in the bounding box and iteratively update their positions by following the vector field until convergence. Converged points are then clustered using DBSCAN to extract atomic coordinates.

**Scalability Analysis.** We analyze the scalability of the proposed field-to-molecule reconstruction procedure from both practical and theoretical perspectives. In practice, we observe that the query point configurations listed in Table 6 are sufficient to achieve stable and accurate reconstructions across datasets. Notably, the total number of query points used for GEOM-Drugs is approximately proportional to that of QM9, consistent with the ratio between their average molecular sizes. This suggests that the required number of query points scales linearly with molecular complexity rather than with dataset-specific heuristics.

While the reconstruction process is conceptually described as initializing query points uniformly in space, in practice we adopt a **lightweight adaptive selection strategy** to allocate query points more efficiently. We first sample a larger pool of candidate points uniformly within the bounding volume, evaluate the predicted vector field at these locations, and assign each candidate a local field variation score based on the variance of field magnitudes among its nearest neighbors in field space. Candidates with higher variation—typically near atomic attraction basins or inter-atomic boundaries—are assigned higher sampling probability via softmax normalization, and the final query points are selected by multinomial sampling. As a result, query points concentrate in regions with informative field geometry, while redundant sampling in flat regions is avoided.

From a theoretical standpoint, assuming that query points sufficiently cover atomic attraction basins, a molecule with $N$ atoms requires at least

$$N \cdot N_{\min} \tag{20}$$

converged particles to ensure that each atomic center forms a valid density cluster under DBSCAN. The adaptive selection

described above improves the efficiency of this coverage without altering the underlying convergence behavior.

The number of gradient ascent iterations $n_{\text{iter}}$ is set conservatively and is not a computational bottleneck in practice, as particles typically converge well before reaching the maximum iteration count. As a result, the overall computational cost of the reconstruction procedure is dominated by field queries and scales linearly with the number of selected query points. Since the number of query points itself grows approximately linearly with the number of atoms, the proposed reconstruction algorithm exhibits

$$O(N) \tag{21}$$

time complexity with respect to molecular size.

## B. Design and Analysis of the Field Representation

### B.1. Geometric Intuition of Vector-Field–Guided Reconstruction

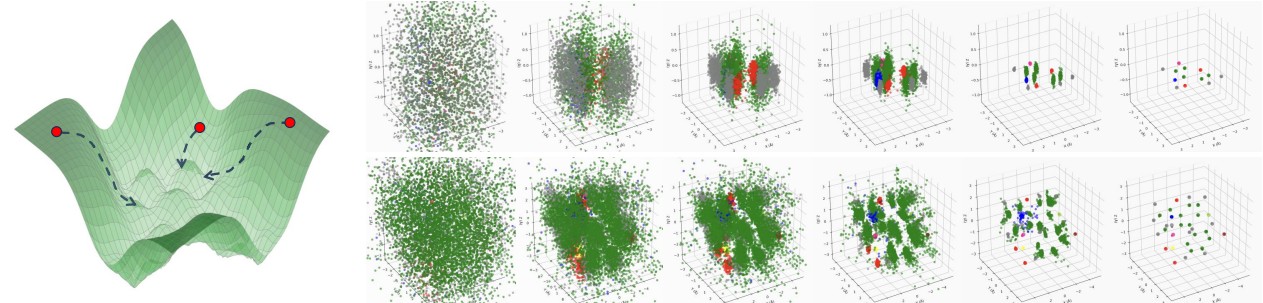

*Figure 7.* **Geometric intuition of gradient-field–guided molecule reconstruction. Left:** An intuitive analogy between the predicted gradient vector field and a conceptual energy-like landscape. The scalar potential itself is not explicitly modeled and may not exist globally; the visualization serves only to illustrate the geometry induced by the learned vector field. Particles (red dots) follow local vector directions and move along curved trajectories toward stable equilibrium regions, which correspond to atomic centers. **Right:** Evolution of particle clustering during field-to-molecule reconstruction. Particles are initialized uniformly within a bounding box and iteratively updated by following the predicted vector field. Over successive iterations, particles converge toward stable equilibrium regions and form dense clusters. Final atomic coordinates are recovered by applying DBSCAN to the converged particle set, with different colors indicating different atom types.

A useful analogy for understanding our reconstruction process is to consider a smooth energy landscape with multiple basins of attraction, as illustrated in Fig. 7. If a large number of particles are released above such a surface, they would follow downhill directions and eventually settle at stable equilibrium points located at the bottoms of the basins.

This analogy is introduced purely for geometric intuition. In our method, the scalar potential surface itself is never explicitly defined or reconstructed, and the learned vector field is not required to be conservative or integrable. As a result, a globally consistent energy function may not exist. Instead, we directly model a three-dimensional vector field that plays the role of a gradient, specifying at each spatial location in $\mathbb{R}^3$ a local direction that guides particles toward nearby stable equilibrium regions.

From this perspective, the illustrated potential landscape should be understood as a conceptual visualization rather than a formal representation. In practice, we directly operate on the vector field: the learned neural field outputs a vector at each spatial location, indicating the local direction and magnitude of movement toward regions corresponding to atomic centers.

### B.2. Scalar vs. Vector Field Representation

To isolate the benefit of directional vector fields over scalar-valued alternatives, we conduct a controlled ablation on a 4,000-molecule QM9 test subset. Using the exact same neural field architecture, latent grid $L^3$, and backbone, changing only the field head, we compare our vector-field prediction against a scalar occupancy head with peak decoding (analogous to FuncMol). Results are shown in Table 7.

The vector-field approach achieves significantly higher geometric precision (RMSD 0.0541 Å vs. 0.1490 Å), demonstrating that the directional information encoded in our vector field is the primary driver of reconstruction accuracy, rather than other architectural choices.

*Table 7.* Controlled ablation: vector field vs. scalar occupancy, same architecture and backbone.

| Field head | Success (%) | Mean RMSD (Å) | AE params |
|---|---|---|---|
| Vector (NF) [Ours] | 99.23 | 0.0541 | $\approx$10.2M |
| Scalar occupancy + peak decode | 99.10 | 0.1490 | $\approx$10.2M |

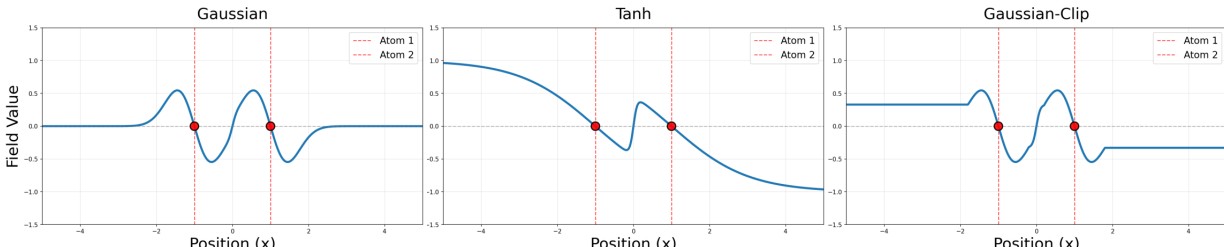

*Figure 8.* **One-dimensional illustration of different field definitions.** Two atoms are placed on a line, and the induced field values along the spatial axis are visualized. Red markers indicate atomic positions.

## B.3. Candidate Field Definitions and Type-Exclusivity

**Design Objective: A Physically Plausible and Learnable Field.** The goal of field–molecule conversion is to define a continuous vector field whose induced dynamics can reliably guide particles toward stable equilibrium regions corresponding to atomic coordinates. From a physical perspective, the field should resemble an energy-like landscape with well-localized attractors, such that particles initialized in the vicinity of an atom are predominantly influenced by that atom and converge unambiguously to its location. This locality is essential to ensure precise reconstruction of individual atomic centers without interference from nearby atoms.

From a learning perspective, however, the vector field must also satisfy several practical constraints. First, the field should vary smoothly across space, avoiding abrupt transitions or discontinuities that are difficult for neural networks to approximate and may lead to unstable optimization. Second, the magnitude of the field must remain bounded and well-conditioned, particularly near atomic centers, as unbounded or rapidly diverging vectors can cause numerical instabilities during training. Finally, the field should retain informative directional signals at moderate to large distances from atoms, so that particles initialized far from any atomic center can still receive meaningful guidance and efficiently move toward equilibrium regions.

In practice, these objectives are often in tension. Fields with extremely sharp, localized attractors provide strong physical intuition but tend to produce vanishing gradients in most of space, making them difficult to learn and inefficient for particle-based reconstruction. Conversely, overly smooth or saturated fields may be easier to optimize but risk introducing ambiguous attraction patterns, spurious equilibria, or unstable attention across different atom types. In this section, we describe how we progressively refined the field definition to balance physical interpretability and learnability, culminating in the final Gaussian-Clip field used in our model.

**Candidate Field Definitions.** We consider three representative field constructions that illustrate the design evolution, and analyze their qualitative behaviors in a simplified one-dimensional setting where atoms are placed at fixed locations along a line.

**Gaussian Field.** A natural choice is to associate each atom with an isotropic Gaussian potential and define the field as the gradient of the resulting energy surface. This construction yields a physically intuitive landscape with smooth, radially symmetric attractors centered at atomic locations. However, the Gaussian field decays rapidly with distance, leading to near-zero gradients in large regions of space. This behavior is illustrated in Figure 8 (left). As a result, the network receives limited learning signal for particles far from atoms, which negatively impacts training stability and convergence.

**Tanh Field.** To mitigate gradient vanishing, one may replace the Gaussian kernel with a tanh-based potential. This modification yields bounded gradients over a wider spatial range and improves numerical stability during training. However, the tanh field saturates at large distances, producing broad and weakly localized attraction regions (Figure 8, middle), which blur the distinction between neighboring atoms. In practice, this can cause particles to drift toward incorrect atom types or intermediate regions, reducing reconstruction precision.

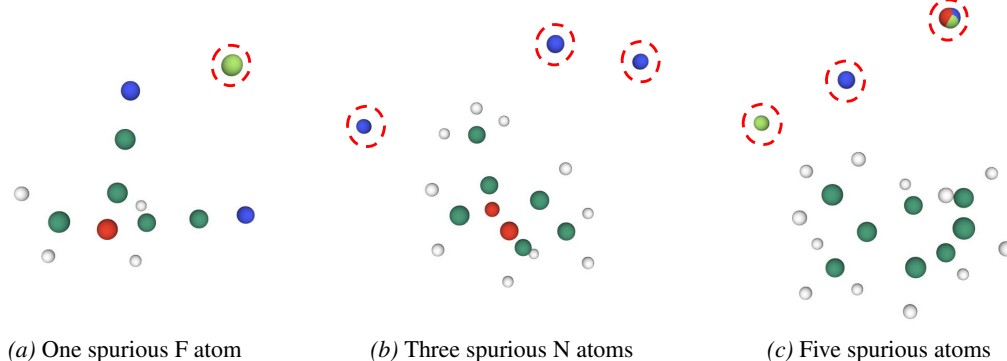

*(a)* One spurious F atom       *(b)* Three spurious N atoms       *(c)* Five spurious atoms

*Figure 9.* **Failure cases without type-exclusive field components.** Examples of reconstructed molecules exhibiting spurious atoms when type-exclusive field components are removed. Light green, blue, and red spheres denote F, N, and O atoms, respectively. Atoms circled by dashed lines correspond to spurious atoms that do not exist in the ground-truth molecules but are incorrectly generated during field-to-atom reconstruction.

**Gaussian-Clip Field.** To combine the advantages of both approaches, we introduce the Gaussian-Clip field. Specifically, the Gaussian potential is smoothly clipped to enforce a lower bound on its magnitude while preserving its localized structure near atomic centers. This construction maintains meaningful gradients even at moderate distances, while avoiding the excessive saturation effects observed in $\tanh$-based fields. As illustrated in Figure 8 (right), the Gaussian-Clip field preserves localized structure near atomic centers while maintaining informative gradients over a wider spatial range. Empirically, this yields more stable particle dynamics and more accurate convergence to atomic equilibria.

**Type-Exclusive Field Design for Closed Atom-Type Spaces.** The final field design enforces type exclusivity by explicitly modeling both attractive and repulsive contributions across the atom type space. For atom types absent from a given molecule, the corresponding field components are constructed to generate outward-pointing vectors, preventing particles from forming spurious equilibria associated with nonexistent atom types.

Formally, let $\mathbf{c} = \frac{1}{N}\sum_{i=1}^{N}\mathbf{x}_i$ be the geometric center (centroid) of all atoms present in the molecule. For an absent atom type $k$, the repulsive field at query point $\mathbf{q}$ is defined as:

$$\mathbf{v}_k^*(\mathbf{q}) = w_{\text{rep}}(\mathbf{q}) \cdot \frac{\mathbf{q} - \mathbf{c}}{\|\mathbf{q} - \mathbf{c}\| + \epsilon}, \tag{22}$$

where the direction strictly points outward from the molecular center. The magnitude term $w_{\text{rep}}(\mathbf{q})$ is given by:

$$w_{\text{rep}}(\mathbf{q}) = \gamma \cdot \exp\left(-\frac{\|\mathbf{q} - \mathbf{c}\|^2}{2\sigma_{\text{rep}}^2}\right), \tag{23}$$

with $\gamma > 0$ controlling the repulsion strength and $\sigma_{\text{rep}}$ scaled according to the molecule's spatial extent. This ensures repulsive vectors are strong near the molecular core (where spurious atoms might erroneously form) and smoothly vanish at a distance.

This design ensures that the field provides informative signals for both presence and absence, resulting in a closed and consistent formulation over the full atom type space. All contributions are integrated directly into the field definition, without relying on auxiliary masking or post-hoc heuristics. Empirically, this formulation improves reconstruction robustness, particularly for molecules with sparse or imbalanced atom type distributions.

The necessity of enforcing type exclusivity is further illustrated in Fig. 9. Without explicit repulsive field components for absent atom types, the learned field may still admit locally stable equilibria that do not correspond to any valid atomic center. As a result, particles can collapse into spurious attractors, leading to the emergence of ghost atoms, incorrect atom types, or merged clusters during reconstruction.

These failure cases are not isolated corner cases, but systematic behaviors observed across different molecules when type exclusivity is not enforced. By explicitly encoding absence through repulsive field contributions, the final field definition effectively suppresses such spurious equilibria and yields more stable and semantically consistent reconstructions.

## B.4. Latent Space PCA Analysis

To assess the structure of the learned latent space, we perform linear PCA on the latent codes of 5,000 QM9 test molecules. The cumulative variance explained by the top principal components is:

*Table 8.* Cumulative variance explained by principal components of the QM9 latent space.

| Number of PCs | Cumulative variance (%) |
|---|---|
| 1 | 6.78 |
| 10 | 35.0 |
| 32 | 52.8 |
| 80 | 66.7 |

Variance is spread across many directions, indicating that the latent space is not redundant and that linear dimensionality reduction alone does not fully capture the intrinsic structure of the neural field codes.

# C. Comprehensive Ablation Studies

## C.1. Autoencoder Capacity and Grid Resolution

**Grid Resolution.** We ablate the effect of latent grid resolution $L$ on reconstruction quality. For different $L$, the grid spacing is adjusted such that all grids cover the same 3D spatial extent (cube edge length = 12 Å), isolating the effect of latent resolution without confounding differences in spatial coverage.

*Table 9.* Reconstruction quality vs. latent grid resolution $L$ on QM9, with fixed spatial extent.

| $L^3$ grid cells | AE hidden 384 | | AE hidden 1024 | |
|---|---|---|---|---|
| | Success (%) | Mean RMSD (Å) | Success (%) | Mean RMSD (Å) |
| $5^3 = 125$ | 99.2 | 0.054 | 99.5 | 0.031 |
| $4^3 = 64$ | 81.3 | 0.107 | 87.2 | 0.083 |
| $3^3 = 27$ | 64.9 | 0.204 | 71.8 | 0.120 |

Reconstruction quality improves significantly with higher grid resolution for both model capacities, confirming that the latent codes encode spatially informative structure and that the latent space is not redundant.

**Autoencoder Capacity on GEOM-Drugs.** To assess the impact of autoencoder capacity on downstream generation quality, we compare two configurations on GEOM-drugs: the default MLP hidden width of 384 and a larger width of 1024 (both with grid size $L = 7$ and latent dimension $d = 384$). Results are reported in Table 10.

*Table 10.* Generation results on GEOM-drugs with different autoencoder hidden widths. Means and standard deviations are reported over 3 independent runs.

| | mol% ↑ | atom% ↑ | valid ↑ | unique ↑ | val. $W_1$ ↓ | aTV ↓ | bTV ↓ | b.len $W_1$ ↓ | b.ang $W_1$ ↓ | frag ↑ | strain ↓ | r.TV ↓ | QED ↑ | SA ↑ | logP ↑ |
|---|---|---|---|---|---|---|---|---|---|---|---|---|---|---|---|
| H384 | 77.5 (±0.3) | 98.9 (±0.1) | 82.5 (±0.3) | 80.9 (±0.5) | 0.099 (±0.004) | 0.033 (±0.001) | 0.272 (±0.002) | 0.083 (±0.020) | 8.44 (±0.22) | 67.27 (±0.64) | 58.1 (±2.4) | 0.434 (±0.065) | 0.639 (±0.000) | 2.66 (±0.25) | 2.18 (±0.16) |
| H1024 | 80.2 (±0.3) | 99.1 (±0.1) | 89.2 (±0.4) | 91.2 (±0.3) | 0.108 (±0.003) | 0.034 (±0.000) | 0.112 (±0.001) | 0.024 (±0.001) | 4.81 (±0.20) | 81.82 (±0.95) | 87.3 (±3.4) | 0.296 (±0.037) | 0.653 (±0.001) | 2.08 (±0.19) | 2.30 (±0.17) |

Notably, increasing autoencoder capacity consistently improves nearly all metrics, with particularly large gains in validity (+6.7 pp), uniqueness (+10.3 pp), bond-level geometry (bond length $W_1$ from 0.083 to 0.024; bond angle $W_1$ from 8.44 to 4.81), and bond topology (Bonds TV from 0.272 to 0.112). Together with the grid resolution results above, these findings confirm that reconstruction fidelity, rather than the latent representation or the diffusion process, is the primary bottleneck, and that scaling model capacity is an effective path to improving generation quality on complex molecular distributions.

## C.2. Field Definition Parameters

**Quantitative Comparison of Field Definitions.**  We quantitatively compare three field definitions on the QM9 dataset, using a held-out test set of 4,000 molecules. For each molecule, particles are initialized uniformly in space and evolved under the predicted vector field, after which atomic coordinates are recovered via the same reconstruction procedure. We evaluate reconstruction quality using both the success rate and the root-mean-square deviation (RMSD) between reconstructed and ground-truth atomic positions.

To disentangle the intrinsic geometric properties of different field formulations from errors introduced by the learned neural field, we conduct experiments using two data sources: ground-truth fields computed directly from molecular structures, and predicted fields decoded from learned latent codes. The quantitative results are summarized in Table 11. RMSD is omitted for ground-truth fields because empirical measurements show that the resulting RMSD is on the order of $10^{-9}$, effectively at numerical precision, making quantitative comparison uninformative.

*Table 11.* Quantitative comparison of different field definitions on the QM9 test set.

| Field Definition | Data Source | Success Rate (%) | RMSD (Å) |
|---|---|---|---|
| Gaussian | Ground Truth | 100 | – |
| Gaussian | Codes | 70.67 | 0.106829 |
| Tanh | Ground Truth | 100 | – |
| Tanh | Codes | 94.13 | 0.094145 |
| Gaussian-Clip | Ground Truth | 100 | – |
| Gaussian-Clip | Codes | 96.75 | 0.055479 |
| Gaussian-Clip with Exclusive Field | Ground Truth | 100 | – |
| Gaussian-Clip with Exclusive Field | Codes | 99.23 | 0.054073 |

For efficiency, all field definition comparisons are conducted using a lightweight neural field autoencoder with 10.2M parameters, approximately one-third the size of the model used in the main experiments.

The results show that the Gaussian field suffers from reduced reconstruction reliability, while the Tanh field exhibits degraded spatial precision. In contrast, the Gaussian-Clip field consistently achieves the best overall performance across both data sources and evaluation metrics, indicating a favorable trade-off between physical localization and learnability. These findings justify our choice of Gaussian-Clip as the default field definition in subsequent experiments.

**Parameter Choices for the Gaussian-Clip Field.**  We perform a systematic exploration of the key parameters defining the Gaussian-Clip field, namely $\sigma_{\mathrm{sf}}$, $\sigma_{\mathrm{mag}}$, and $d_{\mathrm{clip}}$. Here, $\sigma_{\mathrm{sf}}$ and $\sigma_{\mathrm{mag}}$ control the overall shape and smoothness of the Gaussian curve, determining how sharply the field varies near atomic centers. The clipping distance $d_{\mathrm{clip}}$ defines the radius beyond which the gradient magnitude is truncated.

During parameter tuning, we observed the following trends:

- When $d_{\mathrm{clip}}$ is too small, the gradient field becomes locally very steep, making it difficult for the neural network to learn stable and smooth updates.

- When $d_{\mathrm{clip}}$ is too large, the gradients at the clipping distance decay to zero, which undermines our original design goal of maintaining meaningful long-range interactions.

- $\sigma_{\mathrm{sf}}$ and $\sigma_{\mathrm{mag}}$ primarily shape the local curvature and amplitude of the Gaussian; improper choices can lead to either overly diffuse or excessively sharp fields.

After extensive empirical evaluation, we select the combination

$$\sigma_{\mathrm{sf}} = 0.1, \quad \sigma_{\mathrm{mag}} = 0.45, \quad d_{\mathrm{clip}} = 0.8, \tag{24}$$

which provides a smooth yet informative field suitable for neural learning, balancing local sharpness and long-range consistency.

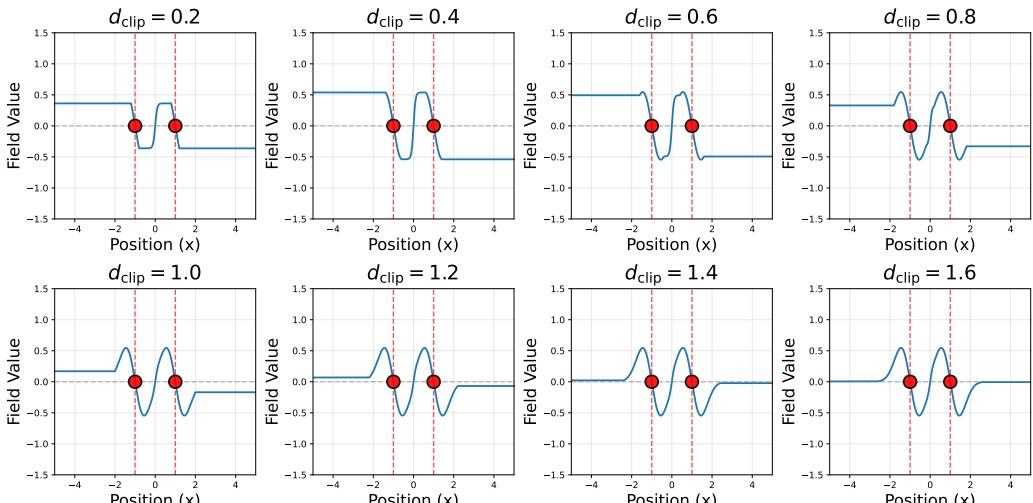

*Figure 10.* Ablation study of the Gaussian-Clip field parameters. We fix $\sigma_{\text{sf}}$ and $\sigma_{\text{mag}}$ to control the general shape of the Gaussian, and vary $d_{\text{clip}}$ to examine its effect on the gradient field. Small $d_{\text{clip}}$ values create steep local gradients that are hard to learn, while large $d_{\text{clip}}$ values suppress long-range gradients. The selected combination ($\sigma_{\text{sf}} = 0.1, \sigma_{\text{mag}} = 0.45, d_{\text{clip}} = 0.8$) achieves a balance between smoothness and long-range interactions.

## C.3. Reconstruction Hyperparameters

Figure 11 analyzes the effect of the DBSCAN radius $\varepsilon$ and the number of particle evolution iterations $n_{\text{iter}}$ on the QM9 test set. We observe that varying $\varepsilon$ over a wide range has negligible impact on both reconstruction success rate and RMSD. This indicates that particles associated with the same atomic basin are already tightly localized, so that changing the clustering radius does not alter cluster assignments or centroids. In this regime, RMSD is primarily governed by the accuracy of particle convergence rather than the post-processing radius.

This observation also suggests that the commonly used stability condition $\|\mathbf{g}(\mathbf{x})\| \cdot \eta < \varepsilon$ should be interpreted as a transient guideline during early particle evolution. Near atomic centers, the predicted field magnitude $\|\mathbf{g}(\mathbf{x})\|$ rapidly approaches zero, causing particle displacements to vanish regardless of the step size. As a result, cluster compactness is dominated by the converged field geometry rather than the explicit choice of $\varepsilon$.

In contrast, the number of evolution iterations $n_{\text{iter}}$ has a clear effect on reconstruction quality. As $n_{\text{iter}}$ increases, particles progressively settle into equilibrium regions, leading to monotonic improvements in success rate and RMSD until convergence is reached. Beyond this point, additional iterations yield diminishing returns. To ensure reliable convergence across molecules of varying complexity, we adopt $n_{\text{iter}} = 500$ in all experiments.

Based on this robustness analysis, we use a slightly larger clustering radius $\varepsilon = 0.1$ to increase tolerance to residual noise without affecting reconstruction accuracy. All remaining reconstruction parameters are fixed as summarized in Table 6.

## C.4. Diffusion Neighborhood Radius

Latent field anchor points preserve spatial structure, so it is desirable for the EGNN to respect locality. We construct edges using a radius threshold $r$, forming the edge set

$$\mathcal{E}_r = \{(i, j) \mid \|\mathbf{x}_i - \mathbf{x}_j\| \leq r\}. \tag{25}$$

We performed an ablation study on the neighborhood radius $r$ using the QM9 validation set. The validation loss is defined as the mean squared error between the clean latent codes $\mathbf{z}_0$ and denoised predictions $\hat{\mathbf{z}}_0$. Figure 12 summarizes the results.

Results indicate a trade-off associated with the radius parameter. Smaller radii tend to produce more fragmented graphs and slightly higher residual errors, while larger radii may reduce the locality inductive bias. Intermediate values (e.g., $r/\text{anchor} = 1.8$) appear to offer a reasonable balance and are adopted for all experiments. This parameter shows relative robustness across a range of values and is expected to generalize to other molecular datasets, as it primarily influences latent code connectivity rather than the absolute molecular scale.

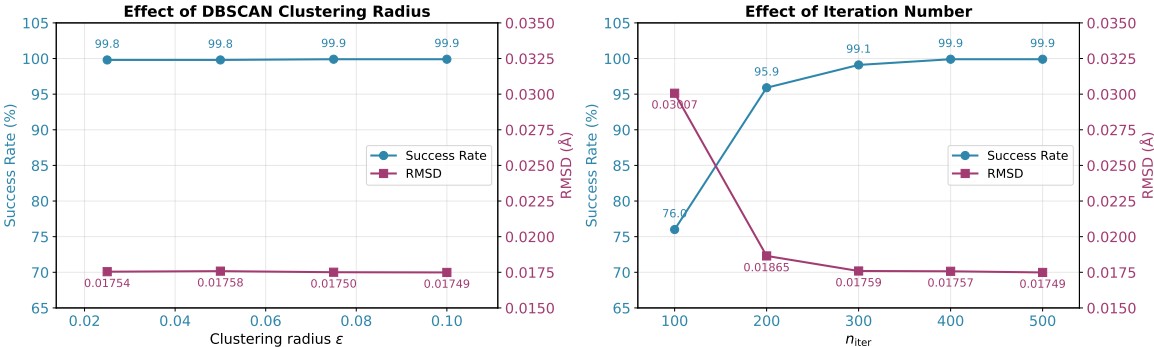

*Figure 11.* **Robustness of density-based atom extraction to post-processing parameters.** Left: Reconstruction success rate and RMSD as functions of the DBSCAN clustering radius $\varepsilon$ on the QM9 test set. Performance remains nearly unchanged across a wide range of $\varepsilon$, indicating that particles are already tightly localized around atomic centers after convergence. Right: Effect of the number of particle evolution iterations $n_{\text{iter}}$. Increasing $n_{\text{iter}}$ improves reconstruction quality until convergence, after which gains saturate. Based on this analysis, we use $\varepsilon = 0.1$ and $n_{\text{iter}} = 500$ in all experiments.

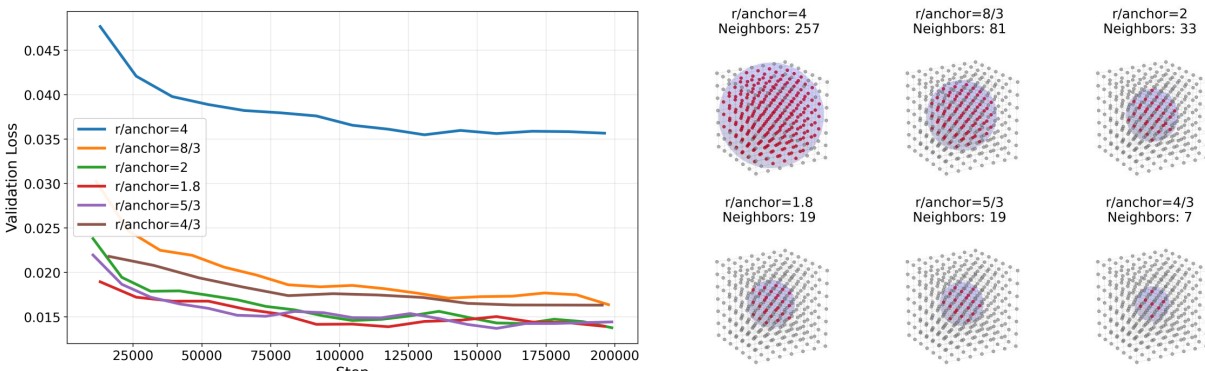

*Figure 12.* **EGNN radius ablation for latent diffusion.** *Left:* Validation loss (MSE between clean and denoised latent codes) over training steps for different neighborhood radii. Intermediate radii yield the most stable optimization and lowest residual error. *Right:* Illustration of radius graphs and number of neighbors per anchor. Small radii produce sparse connectivity, limiting message passing, while overly large radii weaken locality bias. The optimal radius balances connectivity and local geometric structure.

# D. Extended Evaluation and Efficiency

## D.1. Continuous Reconstruction Metrics

To complement our binary reconstruction success metric (which requires exact matches in atom counts and element types), we report two continuous metrics under a unified evaluation protocol across QM9, GEOM-drugs, and CREMP.

**Element-level matching (`atom_match_ratio`).** After type-aware optimal matching via the Hungarian algorithm (with infinite cost for element mismatches), we measure the fraction of ground-truth atoms that receive a same-element partner in the reconstructed set. This score remains in $[0, 1]$ and stays informative even when total atom counts disagree.

**Coordinate-level RMSD (`rmsd_kabsch`).** On the same-element matched pairs, we report the RMSD after Kabsch alignment (optimal rotation and translation), providing a fine-grained, continuous measure of geometric agreement that is invariant to rigid motions.

## D.2. Sampling Time Breakdown

All sampling times are measured on a single NVIDIA RTX 4090 GPU using each method's released code and official checkpoints where available. N = 10,000 molecules are generated per method. For VecMol, the latent diffusion batch size is 1000, and timings are averaged over 3 independent runs; for baseline methods (EDM, GeoLDM, VoxMol), timings are averaged over 10 repeats.

*Table 12.* Continuous reconstruction metrics across datasets. `atom_match_ratio` measures element-level correspondence; `rmsd_kabsch` measures coordinate-level error after Kabsch alignment.

| Dataset | $N_{\text{eval}}$ | atom_match_ratio | | | Dataset | rmsd_kabsch (Å) | | | |
|---|---|---|---|---|---|---|---|---|---|
| | | Mean | St. dev. | Median | | Mean | St. dev. | Median | Q25–Q75 |
| QM9 | 13,388 | 0.996 | 0.017 | 1.000 | QM9 | 0.0356 | 0.0151 | 0.0345 | 0.0309–0.0387 |
| GEOM-drugs | 10,000 | 0.995 | 0.019 | 1.000 | GEOM-drugs | 0.0504 | 0.0319 | 0.0463 | 0.0377–0.0563 |
| CREMP | 3,619 | 0.995 | 0.013 | 1.000 | CREMP | 0.0398 | 0.0206 | 0.0277 | 0.0218–0.0342 |

*Table 13.* Stage-wise sampling time breakdown for VECMOL (in seconds, batch size = 1000).

| Stage | QM9 (s) | GEOM-drugs (s) |
|---|---|---|
| Diffusion (DDPM) | 6.92 | 7.75 |
| Gradient ascent | 8.01 | 23.84 |
| DBSCAN | 0.01 | 0.04 |
| Total | 14.94 | 31.62 |

As the molecule size scales up, the main bottleneck shifts from the diffusion process itself to the downstream continuous optimization (gradient ascent).

