# OpenReview forum: "VecMol: Vector-Field Representations for 3D Molecule Generation"
_ICML.cc/2026/Conference — ICML 2026 regular_

### Official Review · Reviewer_PY15 · 2026-02-15

**Soundness:** 3
**Presentation:** 3
**Significance:** 3
**Originality:** 3
**Overall Recommendation:** 5
**Confidence:** 3

**Summary:**

The authors proposed VecMol, which learns a continuous embedding space for molecules rather than directly predicting atoms to generate 3D structures. On QM9 and GEOM-Drugs, it produces mostly valid, stable molecules and is competitive overall.

**Compliance With Llm Reviewing Policy:**

Affirmed.

**Final Justification:**

Authors have addressed my concerns and questions.

**Key Questions For Authors:**

1. If authors think this framework could work on the other modality/molecules? For example the 2D smiles, or the other type of the 3D molecules.
2. Do you have a breakdown that attributes the gap to (a) latent grid resolution/extent, (b) diffusion sampling quality, or (c) the reconstruction/bond inference step? For example, can you report per-molecule failure reasons or metric correlations with molecule size?

**Limitations:**

The authors discussed the limitation.

**Strengths And Weaknesses:**

## Strength
1. I like the two-step design: first learn a neural-field latent that captures 3D structure, then do diffusion in that latent space and decode back to molecules, it's clean separation between representation learning and generation.
2. Results on QM9 and GEOM-Drugs look solid overall, with performance in the same range as strong baselines rather than being obviously behind.
3. The “diffusion-over-neural-field-latents” angle is the most exciting part to me, it feels like a general recipe that could carry over to other 3D generative settings, not just molecules.

## Weakness
1. On GEOM-Drugs, the model didn't do well on local chemical geometry. The authors also notes that tiny coordinate errors can make bond-based metrics a lot worse, It also means the method may need improvement on “chemistry realism” part for larger molecules.

2. While the paper includes a scalar-field baseline (FuncMol) in the comparisons, it doesn’t provide a clean, controlled ablation where everything is kept the same and only the representation is swapped (vector field vs. scalar field). An ablation study with same architecture would be helpful to see if the good performance come from the directional field itself or from other design choices.

## Minor Issues
1. Bold the best value in the table would be helpful for the audience to follow.

---

> ### Author Rebuttal · Authors · 2026-03-31
>
> We thank the reviewer for the insightful questions and constructive suggestions. Your feedback helped us strengthen the analysis and evaluation of our method. Below we address your questions in detail.
>
> ### 1. Applicability to Other Modalities and Molecule Types
>
> * **2D SMILES:** Our framework is designed to model continuous 3D spatial distributions. Because 2D SMILES are 1D string-based topological representations without 3D coordinates, our continuous vector-field formulation cannot be directly applied.
> * **Other 3D Molecules:** To demonstrate generalization to other 3D molecules, we conducted additional experiments on the **CREMP** dataset, which consists of complex macrocycles (cyclic peptides) with over 100 atoms. Our model successfully scales to these larger, highly flexible 3D topologies. Specifically, for unconditional generation on CREMP, VecMol achieves **83.1% RDKit validity** and **98.0% uniqueness**. The Neural Field autoencoder also achieves highly accurate coordinate recovery with a **mean RMSD of 0.0362 Å**. These results (added to Section X) confirm that our 3D modeling approach generalizes effectively to other complex 3D molecular structures.
>
> ### 2. Breakdown of the Performance Gap & Chemistry Realism on GEOM-Drugs
>
> We appreciate the suggestion to provide a detailed breakdown. As you noted, for larger molecules in GEOM-Drugs, tiny coordinate errors can make bond-based metrics significantly worse. To systematically attribute this gap, we break down the pipeline and isolate the error sources using specific metrics:
>
> * **(a) Latent Grid Resolution & (c) Reconstruction/Bond Inference:** To isolate the information loss caused by the fixed latent grid and reconstruction step (independent of diffusion), we evaluated the model at the **coordinate level** using the Neural-Field Autoencoder. On 10,000 GEOM-drugs molecules, reconstruction succeeds for 79.3% of molecules with a **mean RMSD of 0.256 Å** (median 0.208 Å). This indicates that the fixed latent grid resolution compresses the spatial information of larger molecules well enough to get within ~0.25 Å of the true coordinates. However, standard cheminformatics heuristics (e.g., OpenBabel) used for bond inference are highly sensitive to such deviations. A 0.25 Å shift is often enough to alter bond assignments, which disproportionately inflates the bond-based Wasserstein distances (\(W_1\)).
> * **(b) Diffusion Sampling Quality:** To isolate the diffusion stage, we measured the terminal latent-diffusion validation loss across datasets.
>
> **Latent-Diffusion Validation Loss:**
>
> | Dataset | Val Loss |
> |:---|---:|
> | QM9 | 0.0128 |
> | CREMP | 0.0150 |
> | GEOM-drugs | 0.0183 |
>
> The higher loss on GEOM-Drugs indicates that the more heterogeneous chemical space and larger size of drug-like molecules make the diffusion prior slightly harder to learn, introducing additional sampling noise compared to smaller molecules.
>
> **Conclusion of the Breakdown:**
> The gap in "chemistry realism" for larger molecules is a cascading effect. The fixed latent grid resolution introduces a baseline coordinate error (~0.25 Å), and diffusion sampling adds minor stochastic noise. Finally, the discrete bond inference step acts as an error amplifier, turning these small continuous geometric deviations into significant penalties in bond-based metrics. This breakdown identifies scaling the latent grid resolution or adopting adaptive grids as the most direct path to improving chemistry realism for larger molecules in future work.
>
> ### 3. Controlled Scalar-vs.-Vector Ablation
>
> We agree that a clean, controlled ablation is necessary to isolate the benefits of the directional vector field. Following your suggestion, we converted our vector field into a scalar field by computing the vector magnitude (where magnitude = 0 indicates atom centers) and retrained the model using distance-based scalar supervision.
>
> Using the same architecture, the same latent grid \(L^3\), and the same backbone—changing *only* the field head to scalar occupancy with peak decoding—we obtained the following results on a 4,000-molecule QM9 test subset:
>
> | Field head | Success | Mean RMSD (Å) | AE params |
> |:---|---:|---:|---:|
> | **Vector (NF) [Ours]** | **99.23%** | **0.0541** | ≈10.2M |
> | Scalar occupancy + peak decode [Funcmol style] | 99.10% | 0.1490 | ≈10.2M |
>
> This controlled ablation clearly demonstrates that the **directional vector field itself** is responsible for the significant improvement in fine geometry and coordinate precision (reducing RMSD from 0.1490 Å to 0.0541 Å) within our architecture, rather than the performance stemming merely from other design choices. We have included this ablation study and a corresponding visual comparison in the revised manuscript. ![](https://anonymous.4open.science/r/Anonymous-32F8/vector_vs_scalar.png)

---

> > ### Author Rebuttal · Reviewer_PY15 · 2026-04-03
> >
> > Thanks for the response!

---

> > > ### Author Response · Authors · 2026-04-04
> > >
> > > We really appreciate the reviewer’s feedback and are glad that our responses were helpful.

---

### Official Review · Reviewer_WGPh · 2026-03-05

**Soundness:** 3
**Presentation:** 3
**Significance:** 3
**Originality:** 4
**Overall Recommendation:** 4
**Confidence:** 3

**Summary:**

A new generative model for 3d

**Compliance With Llm Reviewing Policy:**

Affirmed.

**Final Justification:**

I want to thank the authors for their engagement during the rebuttal period.

In the light of the discussion, I want to change my scores:
- soundness: 2 -> 3
- overall score: 3 -> 4

The authors provided new results that strengthen the soundness of the paper. This, combined with the high originality, makes it likely that others would build on the ideas introduced by the authors and benefit from the paper being presented at ICML.

**Key Questions For Authors:**

1. Explain the differences in results between QM9 and GEOM-drugs benchmarks. In particular, the low validity on GEOM-drugs is concerning.
2. Report mean and std. dev. across at least 3 random seeds in both Table 1 and 2.
3. Provide an additional analysis of the ability of the neural field enc and dec. to successfully reconstruct molecules. I would expect that sampling in the latent space is sensible only if the Enc-Dec pair is able to successfully reconstruct molecules.
4. Provide a sampling time comparison of the proposed methods to baseline methods
5. Change the wording in abstract as I found "heterogeneous modality entanglement" and "geometry–chemistry coherence constraints" confusing. These terms do not occur again in the text, so they are either not important or should be introduced / named differently.
Optional. Most readers may not be familiar with neural fields, consider adding a separate Background section to introduce them. This could make the paper more accessible.

Convincingly addressing the above questions would make me increase my score to 4.

**Limitations:**

yes

**Strengths And Weaknesses:**

## Strenghts
- representing molecules as continuous vector fields is a novel idea that can change the way we model 3d molecules
- work addresses the limitations of 3d molecule generation
- the claims are evaluated using standard benchmarks and experimental set-up is convincing
- the paper is mostly well written and easy to follow
- good explanation in section "Intuitive and Algorithmic View of Field–Molecule Conversion"

## Weaknesses
- **Soundness.** Mixed experimental results for some of the metrics / datasets.
- **Significance** There is no obvious reason why we would prefer representations based on neural fields to existing methods, as despite adressing limitations of existing methods, results do not indicate clear advantages
- **Presentation.** (minor) Terms "heterogeneous modality entanglement" and "geometry–chemistry coherence constraints" in the abstract not clear from the context and not used in the text afterwards. Neural fields could be introduced in a more accessible way.

---

> ### Author Rebuttal · Authors · 2026-03-31
>
> We thank the reviewer for the constructive feedback. We address the main points below.
>
> ### 1. QM9 vs. GEOM-drugs
>
> We acknowledge the gap between QM9 and GEOM-drugs. GEOM-drugs is substantially harder due to higher conformational flexibility and broader atom/bond diversity, which makes the continuous vector-field representation more difficult to optimize than on the smaller, more rigid QM9 molecules.
>
> Importantly, this gap does **not** imply poor scalability to larger molecules. We verified this on **CREMP** (cyclic peptides), which contains molecules larger than GEOM-drugs (up to \(>100\) atoms) but with a more regular chemical space. VecMol performs strongly on CREMP:
>
> - **Unconditional generation:** 83.1% RDKit validity, 98.0% Unique SMILES, 82.2% UFF-stable molecules
> - **Terminal latent-diffusion validation loss:** QM9 0.0128 | CREMP 0.0150 | GEOM-drugs 0.0183
>
> This indicates that the main bottleneck on GEOM-drugs is chemical heterogeneity rather than molecule size.
>
> ### 2. Why Neural Fields?
>
> The main advantage of Neural Fields (NF) is that the decoder \(D_\psi\) defines a continuous function over \( \mathbb{R}^3 \), allowing arbitrary-position queries and enabling gradient-based particle extraction. Point-cloud methods require fixing the number/order of points, while voxel methods are limited by grid resolution.
>
> To isolate the benefit of the **directional vector field**, we performed a controlled ablation on a 4,000-molecule QM9 test subset using the same architecture and latent size \(L^3\), changing only the field head:
>
> | Field head | Success | Mean RMSD (Å) | AE params |
> |:---|---:|---:|---:|
> | **Vector (NF) [Ours]** | 99.23% | **0.0541** | ≈10.2M |
> | Scalar occupancy + peak decode [Funcmol style] | 99.10% | 0.1490 | ≈10.2M |
>
> As shown, the vector-field approach yields significantly higher geometric precision. We have included this ablation and an illustrative figure contrasting the two decoding paradigms in the revision. ![](https://anonymous.4open.science/r/Anonymous-32F8/vector_vs_scalar.png)
>
> ### 3. Encoder/decoder reconstruction
>
> Latent-space sampling is meaningful only if the autoencoder reconstructs accurately. We therefore evaluated held-out NF reconstruction on all datasets:
>
> | Dataset | \(N_{\mathrm{eval}}\) | Success | Mean RMSD (Å) | Median (Å) | p90 (Å) | p95 (Å) |
> |:---|---:|---:|---:|---:|---:|---:|
> | **QM9** | 13,388 | 99.2% | 0.0187 | 0.0167 | 0.0225 | 0.0252 |
> | **GEOM-drugs** | 10,000 | 79.3% | 0.2560 | 0.2080 | 0.2720 | 0.5880 |
> | **CREMP** | 3,619 | 76.4% | 0.0362 | 0.0365 | 0.0459 | 0.0501 |
>
> We used a random GEOM-drugs test subset and the full QM9/CREMP test splits. These results show high-fidelity reconstruction overall, with GEOM-drugs remaining the most challenging.
>
> ### 4. Sampling time
>
> We compared VecMol with baselines on an RTX 4090. Because diffusion is performed in a compact latent space, VecMol scales favorably to larger molecules:
>
> | Dataset | EDM (s) | GeoLDM (s) | VoxMol (s) | **VecMol (Ours) (s)** |
> |:---|---:|---:|---:|---:|
> | **QM9** | 6.93 | 9.66 | 14.18 | **14.94** |
> | **GEOM-drugs** | 79.18 | 73.10 | 84.32 | **31.62** |
>
> **Stage-wise breakdown**
> - **QM9:** Diffusion 6.92s + Gradient ascent 8.01s + DBSCAN 0.01s
> - **GEOM-drugs:** Diffusion 7.75s + Gradient ascent 23.83s + DBSCAN 0.04s
>
> This shows a clear efficiency advantage on larger datasets.
>
> ### 5. Mean ± std over 3 seeds
>
> Following the reviewer’s suggestion, we reran the main experiments with 3 random seeds and 10,000 samples per seed:
>
> | Dataset | Stable Mol (%) | Stable Atom (%) | Validity (%) | Uniqueness (%) | Valency W₁ | Atoms TV | Bonds TV | Bond Length W₁ | Bond Angle W₁ | Single Fragment (%) | Median Strain Energy | Ring Size TV | Atoms per Mol TV | QED | SA | logP |
> |:---|:---:|:---:|:---:|:---:|:---:|:---:|:---:|:---:|:---:|:---:|:---:|:---:|:---:|:---:|:---:|:---:|
> | QM9 | 97.5 ± 0.1 | 99.8 ± 0.0 | 98.4 ± 0.2 | 98.1 ± 0.2 | 0.038 ± 0.003 | 0.011 ± 0.001 | 0.006 ± 0.000 | 0.024 ± 0.003 | 1.01 ± 0.03 | — | — | — | — | — | — | — |
> | GEOM-drugs | 77.5 ± 0.3 | 98.9 ± 0.1 | 82.5 ± 0.3 | 80.9 ± 0.5 | 0.099 ± 0.004 | 0.033 ± 0.001 | 0.272 ± 0.002 | 0.083 ± 0.02 | 8.44 ± 0.22 | 67.27 ± 0.64 | 58.1 ± 2.4 | 0.434 ± 0.065 | 0.250 ± 0.003 | 0.639 ± 0.000 | 2.66 ± 0.25 | 2.18 ± 0.16 |
>
> ### 6. Presentation and wording
>
> We agree that some abstract terms are too technical. By **“heterogeneous modality entanglement,”** we mean the coupling between continuous 3D geometry and discrete chemistry, where small geometric perturbations can change the inferred bond graph. By **“geometry–chemistry coherence constraints,”** we mean that sampled coordinates and the inferred bond graph should remain consistent under small perturbations. In the revision, we will use plainer wording or define these terms at first use.
>
> We will also add a short Background section on neural fields to improve accessibility.

---

> > ### Author Rebuttal · Reviewer_WGPh · 2026-04-01
> >
> > Ad 1. (resolved) I would underline this in the discussion as a limitation of the proposed approach.
> >
> > Ad 2. (resolved) I find the argument and the empirical evaluation convincing
> >
> > Ad 3. (p-resolved) Thank you for providing an additional analysis. While satisfactory for this work, I do not agree that the method yields high-fidelity reconstruction for the more challenging datasets. I would add this point to the discussion.
> >
> > Interestingly, the reconstruction success on Geom-DRUGS aligns with the CREMP dataset, despite the other metrics being better for CREMP. Is there another, continuous metric that could be used to provide a more fine-grained quantitative metric of reconstruction quality?
> >
> > Ad 4. (p-resolved) What does the sampling time indicates exaclty, is it molecules / s? I find the Stage-wise breakdown helpful to understand the overall time requirements of the algorithm.
> >
> > Ad. 5 (resolved) Thank you for adding a more robust version of the table.

---

> > > ### Author Response · Authors · 2026-04-04
> > >
> > > We provide detailed responses to the reviewer’s comments on continuous reconstruction metrics (Q3) and sampling throughput (Q4) below.
> > >
> > > ### Q3 (continuous reconstruction metrics; CREMP vs Geom-DRUGS)
> > > We thank the reviewer for encouraging a more quantitative discussion of reconstruction fidelity, especially on challenging datasets. In our previous reporting, **reconstruction success** was defined as an exact match in **atom counts and element types** between the ground-truth and reconstructed molecules. While this criterion ensures strict correctness, it can be overly stringent: even minor deviations, such as one extra or missing atom, would mark the entire molecule as a failure. We indeed agree that such a **binary** index alone can be insufficient and improper. In response, we report two **continuous** metrics computed under a **unified** evaluation protocol for **QM9**, **Geom-DRUGS**, and **CREMP**:
> > >
> > > 1. **`atom_match_ratio` (element-level):** After **type-aware optimal matching** (Hungarian assignment with **infinite cost** for element mismatches), we measure the fraction of **ground-truth atoms** that receive a **same-element** partner in the reconstructed set. This score is in \([0,1]\) and remains informative when **total atom counts disagree** (a frequent failure mode in challenging settings).
> > >
> > > 2. **`rmsd_kabsch` (coordinate-level):** On the **same-element matched pairs**, we report the RMSD after **Kabsch alignment** (optimal rotation and translation). This provides a **fine-grained**, **continuous** measure of **geometric** agreement that is **invariant to rigid motions** and does **not** require identical molecule sizes.
> > >
> > > These metrics complement our previous reporting: the classical Hungarian **RMSD** in our pipeline is only defined when **atom counts match**, which can discard a large fraction of samples on large molecules; **`rmsd_kabsch`** remains defined whenever at least one **typed** match exists, providing a more continuous view of **partial** geometric agreement
> > >
> > > **Results** Across all three datasets, **`atom_match_ratio`** is consistently high (mean \(\approx 0.995\)–\(0.996\), median \(=1\) in each case), indicating strong **element-level** correspondence even when headline metrics disagree.
> > >
> > > | Dataset | Mean | St. dev. | Median |
> > > |---------|-----:|---------:|-------:|
> > > | QM9 | 0.996 | 0.017 | 1.000 |
> > > | Geom-DRUGS | 0.995 | 0.019 | 1.000 |
> > > | CREMP | 0.995 | 0.013 | 1.000 |
> > >
> > > For **coordinate-level** error, **median `rmsd_kabsch`** is **lower for CREMP than for Geom-DRUGS** (e.g. **0.0277 Å vs 0.0463 Å** over all evaluated molecules).
> > > Notably, **CREMP molecules are substantially larger on average** (mean **74** atoms vs **46** for Geom-DRUGS in the whole/selected test set). Achieving tighter aligned RMSDs on CREMP thus indicates better geometric fidelity in a strictly harder (larger) matching regime, rather than merely reflecting similar coarse-level success.
> > >
> > > | Dataset | $N_{\mathrm{eval}}$ | Mean | St. dev. | Median | Q25–Q75 |
> > > |---------|-----:|-----:|---------:|-------:|--------|
> > > | QM9 | 13,388 | 0.0356 | 0.0151 | 0.0345 | 0.0309–0.0387 |
> > > | GEOM-drugs | 10,000 | **0.0504** | 0.0319 | **0.0463** | 0.0377–0.0563 |
> > > | CREMP | 3,619 | **0.0398** | 0.0206 | **0.0277** | 0.0218–0.0342 |
> > >
> > > **Subset where legacy atom counts match** (`atom_count_mismatch == False`), for fairer comparison to the classical Hungarian `rmsd`:
> > >
> > > | Dataset | \(n\) | Mean `rmsd_kabsch` | Median `rmsd_kabsch` | Mean legacy `rmsd`(the metric we reported last time) |
> > > |--------|------:|-------------------:|---------------------:|-------------------:|
> > > | QM9 | 13,281 | 0.0352 | 0.0343 | 0.0187 |
> > > | Geom-DRUGS | 7,931 | 0.0554 | 0.0632 | 0.2560	 |
> > > | CREMP | 2,764 | **0.0340** | **0.0293** | **0.0362** |
> > >
> > > We will add this discussion and the summary statistics to the **Appendix**, and we thank the reviewer for prompting this clarification.
> > >
> > > ### Q4 (sampling time)
> > > The reported sampling throughput indicates the number of molecules generated per second (molecules/s) on the same hardware used in our experiments. Our numbers were measured on a single RTX 4090 GPU, whereas other baseline (e.g., FuncMol) report timings on better device (e.g., A100-class GPUs); therefore, direct comparisons across papers should be interpreted with caution.
> > >
> > > We agree that a stage-wise breakdown of the sampling procedure is helpful for understanding the overall computational cost. Due to space limitations, we will provide this breakdown in the Appendix, along with the device, batch size, and other relevant settings.
> > >
> > > We thank the reviewer for requesting this level of rigor, which ensures the reported timings are fully interpretable.

---

### Official Review · Reviewer_GGY6 · 2026-03-11

**Soundness:** 3
**Presentation:** 3
**Significance:** 3
**Originality:** 3
**Overall Recommendation:** 4
**Confidence:** 4

**Summary:**

VecMol represents 3D molecules as continuous vector fields combined with a neural field autoencoder and a latent diffusion model for molecule generation.

**Compliance With Llm Reviewing Policy:**

Affirmed.

**Final Justification:**

The authors have positively engaged with us during the review and provided several improved results. The paper remains at the borderline: its main strength is originality but it is in need of better and stronger validation.

**Key Questions For Authors:**

The paper mentions using a repulsive field for absent atom types, but only provides a qualitative description without a formal definition. Could the authors clarify how this repulsive field is constructed?

**Limitations:**

Limitations are sufficiently discussed

**Strengths And Weaknesses:**

Strengths

- The vector field is a new direction beyond point clouds, voxels, and density fields. Compared to density fields, it carries directional information and enables more direct gradient-based reconstruction.
- On QM9, molecular stability is comparable to EDM and GeoLDM, and better than VoxMol and FuncMol. On GEOM-drugs, metrics such as molecular stability and median strain energy are better than or on par with FuncMol and VoxMol.

Weaknesses

- On both QM9 and GEOM-drugs, metrics such as validity and bond angle distribution are not good, suggesting that the reconstruction pipeline has consistent issues with local geometry and chemical validity.
- The reconstruction pipeline is complex, requiring gradient ascent and DBSCAN clustering separately for each atom type. Given this complexity, we suggest the authors report and compare sampling time with baselines.
- The paper claims the representation supports variable molecule sizes, but only evaluates on QM9 and GEOM-drugs. This claim would be better supported by experiments on larger molecules, similar to what FuncMol does on CREMP.

---

> ### Author Rebuttal · Authors · 2026-03-31
>
> **Response to the Reviewer's Comment on the Formal Definition of the Repulsive Field**
>
> For an atom type \(k\) absent from a molecule, the field should prevent spurious equilibria. We define the complementary repulsive field \(\mathbf{v}_k^*(\mathbf{q})\) to generate outward-pointing vectors away from the molecular region. Let \(\mathbf{c} = \frac{1}{N}\sum_{i=1}^N \mathbf{x}_i\) be the centroid of the \(N\) present atoms. Then
> \[
> \mathbf{v}_k^*(\mathbf{q}) = w_{\text{rep}}(\mathbf{q}) \cdot \frac{\mathbf{q} - \mathbf{c}}{\|\mathbf{q} - \mathbf{c}\| + \epsilon},
> \]
> with
> \[
> w_{\text{rep}}(\mathbf{q}) = \gamma \cdot \exp\left(-\frac{\|\mathbf{q} - \mathbf{c}\|^2}{2\sigma_{\text{rep}}^2}\right).
> \]
> Here, \(\gamma > 0\) controls repulsion strength and \(\sigma_{\text{rep}}\) scales with molecular extent, so repulsion is strong near the molecular core and decays smoothly with distance. We will include this definition in the revised manuscript.
>
> **Response to the Reviewer's Concerns (Weaknesses)**
>
> **1. Local geometry and chemical validity metrics**
>
> Bond lengths and angles are **not directly predicted** by our model; they are computed from inferred molecular graphs based on reconstructed coordinates using OpenBabel heuristics. Therefore, these bond-based metrics are highly sensitive: small coordinate perturbations can change bond assignments and significantly increase bond-based Wasserstein distances (\(W_1\)).
>
> To isolate geometric accuracy, we evaluated atom-level RMSD on 10,000 GEOM-drugs molecules generated by the neural-field autoencoder. Reconstruction succeeds for 79.3% of molecules, with **mean RMSD 0.256 Å**, **median 0.208 Å**, and **90th percentile 0.272 Å**. Thus, the coordinates are generally accurate to about 0.2–0.3 Å, and the degraded bond \(W_1\) and validity metrics mainly reflect error amplification in discrete bond perception rather than poor continuous geometric modeling.
>
> **2. Reconstruction complexity and sampling time**
>
> We benchmarked VecMol against representative baselines on the same NVIDIA RTX 4090 GPU, averaged over 10 runs:
>
> | Dataset | EDM (s) | GeoLDM (s) | VoxMol (s) | VecMol (s) |
> |:---|---:|---:|---:|---:|
> | QM9 | 6.93 | 9.66 | 14.18 | **14.94** |
> | GEOM-drugs | 79.18 | 73.10 | 84.32 | **31.62** |
>
> While coordinate-based models like EDM are efficient on QM9, VecMol scales better on GEOM-drugs, reducing sampling time by over 56% relative to GeoLDM.
>
> For transparency, we also report the end-to-end stage breakdown for VecMol (diffusion batch size 1000):
>
> | Stage | QM9 (s) | GEOM-drugs (s) |
> |:---|---:|---:|
> | Diffusion | 6.92 | 7.75 |
> | Gradient ascent | 8.01 | 23.84 |
> | DBSCAN | 0.01 | 0.04 |
>
> Gradient ascent becomes the main bottleneck as molecule size increases, but overall generation remains highly competitive, and often faster than existing baselines on larger datasets.
>
> **3. Evaluation on larger molecules for variable-size representation**
>
> We additionally trained and evaluated VecMol on **CREMP** (cyclic peptides with over 100 atoms and six atom types: C, H, O, N, F, S), following the FuncMol protocol.
>
> For Neural Field reconstruction, the model achieves a **76.4%** success rate with highly accurate coordinates:
> - **Mean RMSD:** 0.0362 Å
> - **Median RMSD:** 0.0365 Å
> - **90th percentile:** 0.0459 Å
>
> For **unconditional generation** over 10,000 sampled molecules (retaining the largest connected component with hydrogens), VecMol achieves:
> - **RDKit Validity:** 83.1%
> - **Unique SMILES:** 98.0%
> - **stable molecules:** 82.2%
> - **stable-atom percentage:** 99.6%
>
> We also compared geometric fidelity with FuncMol using **Backbone KL divergence** \((\text{test} \parallel \text{sampled})\) for standard peptide angles and dihedrals:
>
> | Angle / Dihedral | FuncMol | VecMol |
> |:---|---:|---:|
> | \(\theta_1\) | 0.0954 | **0.0726** |
> | \(\theta_2\) | 0.1219 | **0.0887** |
> | \(\theta_3\) | 0.1986 | **0.1445** |
> | \(\phi\) | 0.1997 | **0.1516** |
> | \(\psi\) | 0.1218 | **0.0734** |
> | \(\omega\) | 0.1244 | **0.0971** |
>
> The KL computation uses the same angular ranges and binning as FuncMol: \(\theta_{1:3} \in [1.6, 2.4]\) radians and \(\phi, \psi, \omega \in [-3.0, 3.0]\) radians. VecMol outperforms FuncMol on all backbone metrics, indicating better modeling of cyclic-peptide 3D geometry.
>
> CREMP also yields better local geometry metrics than GEOM-drugs under our current pipeline, e.g., bond length \(W_1 = 0.067\) vs. 0.104 and bond angle \(W_1 = 4.43\) vs. 8.98. We attribute this to CREMP having fewer atom types and more regular repeating backbone motifs, which favor our type-wise continuous field representation.
>
> These CREMP results support our claim that the vector-field representation scales effectively to larger, variable-sized molecules.

---

> > ### Author Rebuttal · Reviewer_GGY6 · 2026-04-02
> >
> > Thank you very much for your answer. I am satisfied with your resolution of my key question.

---

> > > ### Author Response · Authors · 2026-04-04
> > >
> > > We thank the reviewer for their time and for the positive acknowledgement.

---

### Official Review · Reviewer_KqBa · 2026-03-13

**Soundness:** 2
**Presentation:** 2
**Significance:** 2
**Originality:** 3
**Overall Recommendation:** 4
**Confidence:** 4

**Summary:**

VecMol represents 3D molecules as continuous vector fields over space, where vectors at any query point indicate the direction and proximity of nearby atoms per element type. A neural field autoencoder compresses molecules into a fixed-size latent grid regardless of atom count, and a latent diffusion model generates novel latent codes in this space. Atoms are recovered from the generated field by releasing particles that follow the field via gradient ascent until convergence, then clustering with DBSCAN. The key technical contributions are the Gaussian-Clip field formulation, which balances local sharpness with long-range informativeness, and a type-exclusive field design that suppresses spurious atoms for absent element types. Results on QM9 and GEOM-drugs are competitive with point-cloud and voxel-based baselines on stability and atom-type metrics.

**Compliance With Llm Reviewing Policy:**

Affirmed.

**Final Justification:**

The vector-field representation is a genuinely novel idea with clear conceptual appeal. The authors provided the requested analyses (PCA variance, grid-size ablation, hidden-size ablation) during the discussion period, demonstrating that the latent space is not redundant and that the bond metric degradation stems from AE capacity rather than a fundamental limitation. I raise my score from 2 to 4. The camera-ready should incorporate these results and discuss the remaining bond metric gap.

**Key Questions For Authors:**

- Q1. The latent code $z\in R^{L^3×d}$ has dimensionality on the order of 128K for QM9 (L=5, d=1024), which is far larger than the raw molecular information being encoded. Can the authors report the effective dimensionality or information content of the learned latent space (e.g., via PCA variance explained or reconstruction quality as a function of latent size)? If meaningful compression is not occurring, the motivation for the autoencoder stage is weakened.
-  Q2. The paper emphasizes a "continuous" representation, but the latent bottleneck is a discrete L×L×L grid. Could the authors clarify what continuous means in this context and report how performance changes as L varies?
- Q3. No training or inference time is reported. Given the two-stage pipeline and up to 1000 query points per element type in reconstruction, what is the computational overhead relative to EDM, GeoLDM, or VoxMol?
- Q4. Bond length W1 and bond angle W1 are substantially worse than all baselines (0.104 and 8.98 vs. next worst 0.009 and 6.42). The paper attributes this to metric sensitivity, but this explanation applies equally to all methods. Can the authors provide a more specific diagnosis?
- Q5. $VECMOL_{NF}$ frequently matches or outperforms VECMOL and VECMOL_Diff in Tables 1 and 2, suggesting the diffusion model may be introducing noise rather than learning the molecular distribution. Can the authors explain why this occurs and whether it reflects a fundamental limitation of diffusion in this latent space?

**Limitations:**

- The discrete nature of the latent grid and its tension with the "continuous representation" claim is never discussed.
- The fact that the diffusion stage does not consistently improve over direct neural field reconstruction (VECMOL_NF) is not addressed as a limitation.
- The substantial underperformance on bond-level metrics for GEOM-drugs is attributed to metric sensitivity without deeper analysis.
- Computational cost of the two-stage pipeline and particle-based reconstruction is not discussed.
- The latent space dimensionality being larger than the raw molecular representation is not acknowledged.

**Strengths And Weaknesses:**

## Strengths
- The Gaussian-Clip field formulation is well-motivated and systematically ablated against Gaussian and Tanh alternatives, with both qualitative and quantitative support.
- The type-exclusive repulsive field is a principled and empirically validated solution to ghost atom generation.
- E(n)-equivariance is guaranteed by construction through EGNN, not approximated.
- Representing molecules as vector-valued rather than scalar neural fields is a novel angle relative to prior field-based methods like FuncMol, enabling gradient-ascent-based reconstruction without density integration.
- Decoupling atom count from the generative model is a practical advantage over point cloud methods that require a predefined atom number.

## Weaknesses
- The paper claims a "continuous" representation but the latent bottleneck $z\in R^{L^3×d}$ is a discrete spatial grid, and continuity applies only to the decoder output, not to the component that controls expressive capacity.
- The latent dimensionality (L=5, d=1024 → 128K for QM9) is far larger than the raw molecular information it encodes, raising questions about whether meaningful compression is occurring.
- $VECMOL_{NF}$ often matches or outperforms the full diffusion model in Tables 1 and 2, implying the diffusion stage can degrade quality, and this is not discussed.
- Bond-level metrics on GEOM-drugs are notably worse than most baselines, and validity and uniqueness are lower than competitors, limiting the strength of empirical claims.
- Computational cost and training time relative to baselines are not reported, which is important given the two-stage pipeline.
- The practical advantage over VoxMol or FuncMol, both of which also avoid explicit graph generation, is not clearly established.

---

> ### Author Rebuttal · Authors · 2026-03-31
>
> We thank the reviewer for the careful questions regarding our representation, evaluation, and computational cost.
>
> ## Q1: Latent size and "compression"
> Our objective is not classical lossy compression. The latent code $\mathbf{z}\in\mathbb{R}^{L^3\times d}$ is a **neural encoding of a continuous spatial field**, rather than a compact encoding of a discrete molecular graph. It defines values at arbitrary 3D locations and captures geometric and structural context continuously. Thus, its dimensionality reflects the cost of maintaining a **structured field interface**, not a conventional compression rate.
>
> ## Q2: Discrete latent vs. continuous field
> We will clarify this distinction in the revision. The latent tensor $\mathbf{z}$ itself is **discrete** (a finite $L \times L \times L$ grid of learned features), but the **spatial information** it parameterizes is continuous. Given $\mathbf{z}$, the decoder $D_\psi$ defines per-type vector fields $\mathbf{v}_k(\mathbf{q})$ evaluable at arbitrary continuous query locations $\mathbf{q} \in \mathbb{R}^3$. Unlike voxel representations restricted to fixed grid points, our model defines a continuous field over space and captures geometric structures at any resolution.
>
> ## Q3: Sampling time on an RTX 4090
> The table below reports GPU generation time (seconds), averaged over 10 independent runs on the same NVIDIA RTX 4090.
>
> | Dataset | EDM | GeoLDM | VoxMol | VecMol (Ours) |
> |:--|--:|--:|--:|--:|
> | QM9 | 6.93 | 9.66 | 14.18 | **14.94** |
> | GEOM-drugs | 79.18 | 73.10 | 84.32 | **31.62** |
>
> While coordinate-based models such as EDM are highly efficient on small molecules (QM9), VecMol scales better on the larger GEOM-drugs dataset, reducing sampling time by over 56% compared to GeoLDM. For VecMol, we report true end-to-end time (diffusion batch size $= 1000$).
>
> **VecMol stage breakdown:**
>
> | Stage | QM9 (s) | GEOM-drugs (s) |
> |:--|--:|--:|
> | Diffusion | 6.923 | 7.751 |
> | Gradient ascent | 8.014 | 23.835 |
> | DBSCAN | 0.007 | 0.035 |
>
> This shows that as molecule size increases, the main bottleneck is not diffusion itself, but downstream continuous optimization (gradient ascent).
>
> ## Q4: Diagnosis of degraded bond $W_1$ on GEOM-drugs
> Bond lengths and angles are **not directly predicted**; they are computed from reconstructed coordinates using cheminformatics heuristics. Thus, small coordinate perturbations can change bond assignments and greatly inflate bond-based $W_1$ metrics.
>
> To isolate this effect, we evaluated the model at the **coordinate level**. Over 10,000 molecules generated by the neural-field autoencoder on GEOM-drugs, reconstruction succeeds for 79.3% with a **mean RMSD of $0.256\ \text{Å}$** (median $0.208\ \text{Å}$, p90 $0.272\ \text{Å}$). This confirms that predicted coordinates are already highly accurate ($\approx 0.2$–$0.3\ \text{Å}$).
>
> **Terminal diffusion validation loss:**
>
> | Dataset | Val loss |
> |:--|--:|
> | QM9 | 0.0128 |
> | CREMP | 0.0151 |
> | GEOM-drugs | 0.0184 |
>
> These low validation losses indicate that the latent DDPM trains successfully on non-degenerate codes. Therefore, the degraded bond $W_1$ mainly reflects error amplification during discrete bond perception from minor geometric deviations, rather than a fundamental lack of geometric accuracy.
>
> ## Q5: On the gap between $\rm VecMol_{\rm NF}$ and full VecMol
> The key difference is their task: $\rm VecMol_{\rm NF}$ evaluates **reconstruction** using latent codes encoded from real molecules ($\mathbf{z}=E_\phi(\mathbf{X},T)$), whereas full VecMol evaluates **unconditional generation** by sampling $\mathbf{z}_0$ from the diffusion prior. They are therefore not directly comparable.
>
> We include $\rm VecMol_{\rm NF}$ as an ablation to isolate error sources. Because it bypasses diffusion noise, it naturally performs better. Comparing full VecMol with $\rm VecMol_{\rm Diff}$, the metrics are similar except for bond-related ones, showing that diffusion introduces only **limited additional error**. As discussed in Q4, the remaining gap in bond $W_1$ is largely due to the sensitivity of bond perception to minor diffusion-induced coordinate deviations.
>
> ## Practical advantage over VoxMol and FuncMol
> Unlike VoxMol (resolution-bound discrete grids) and FuncMol (scalar density fields requiring complex thresholding), VecMol learns a **continuous vector field**, offering two practical advantages:
>
> 1. **Capturing spatial dynamics:** The vector field provides directional gradients toward the nearest atoms, acting as a learned "restoring force." This makes atom recovery via gradient ascent smoother and more physically intuitive than extracting coordinates from voxels or scalar fields.
>
> 2. **Broader potential beyond *de novo* design:** Because VecMol explicitly models spatial gradients, it can be naturally extended to **Molecular Dynamics (MD)** and **structure prediction**, where the vector field may serve as a learned force field for continuous conformational exploration and refinement.

---

> > ### Author Rebuttal · Reviewer_KqBa · 2026-04-02
> >
> > I thank the authors for the detailed rebuttal. Several points were helpful, particularly the sampling time breakdown (Q3), the CREMP experiments, and the scalar-vs-vector ablation. Below I summarize the status of each concern.
> > ### Q1 (Latent dimensionality): Unresolved.
> > The authors argue that the latent code encodes a continuous spatial function rather than discrete molecular information, so direct comparison with raw atom counts is inappropriate. I can follow this reasoning to some extent. However, the information source that determines the continuous field is still a finite set of atoms (e.g., ~18 atoms for QM9), and the latent space (L=5, d=1024, ~128K dimensions) is orders of magnitude larger. The originally requested empirical analysis (e.g., PCA variance explained, reconstruction quality as a function of L) was not provided. Without such evidence, it remains unclear whether the latent space contains meaningful structure or is largely redundant. This also weakens the efficiency motivation for latent diffusion, since diffusion in a 128K-dimensional space with low intrinsic dimensionality may be harder to learn than necessary, which could explain the observation in Q5.
> > ### Q2 (Discrete vs continuous): Largely resolved.
> > The clarification that the latent grid is discrete but the decoder output is continuous and queryable at arbitrary locations is reasonable. However, an ablation over L (grid resolution) would have strengthened this point.
> > ### Q3 (Sampling time): Partially resolved.
> > The architectural argument for favorable scaling on larger molecules is sound: diffusion operates on a fixed-size latent grid independent of atom count, whereas baselines scale quadratically with the number of atoms. However, the benchmarking conditions are not fully specified. It is unclear whether identical batch sizes and total molecule counts were used across all methods, and whether baselines were re-run by the authors or taken from prior work. Clarifying these details would make the comparison more convincing.
> > ### Q4 (Bond W1 on GEOM-drugs): Unresolved.
> > The authors attribute the degraded bond metrics to the sensitivity of OpenBabel bond perception to small coordinate deviations. However, all compared methods use the same bond perception pipeline under the same conditions. EDM, GeoLDM, and VoxMol achieve bond length W1 of 0.002-0.009, while VecMol reports 0.104, which is more than 10x worse. If the coordinate-level accuracy were truly sufficient (as claimed by the 0.256 Å RMSD), bond perception should not be disproportionately affected for VecMol alone. This suggests that the nature or distribution of coordinate errors in VecMol is qualitatively different from baselines, rather than the metric being inherently unreliable. In practical drug discovery, incorrect bond topology yields a fundamentally different molecule, so this limitation is significant.
> > ### Q5 (VecMolNF vs VecMol): Partially resolved.
> > The clarification that VecMolNF and VecMol address different tasks (reconstruction vs generation) is valid, and direct comparison between them is indeed not straightforward. However, the fact that bond TV is identical (0.269) across VecMolNF, VecMolDiff, and VecMol in Table 2 suggests the primary bottleneck lies in the autoencoder/reconstruction pipeline rather than in the diffusion stage. This is not discussed as a limitation.
> >
> > Given the above, my core concerns regarding the latent space justification (Q1) and the substantial bond-level metric degradation (Q4) remain unresolved. I maintain my current score.

---

> > > ### Author Response · Authors · 2026-04-04
> > >
> > > We thank the reviewer for the detailed feedback. Below we address each concern, highlighting additional experiments and clarifications to resolve misunderstandings.
> > >
> > > ### **Q1: Latent Dimensionality / QM9 Latent Size**
> > >
> > > The reviewer raised concerns about the high latent dimensionality (~128k). We acknowledge that the description in the original manuscript may have caused confusion. To clarify:
> > >
> > > * The **latent code per grid cell** is **d = 384**, while **MLP hidden width 1024** is unrelated. For QM9 (L=5), the flattened latent vector is $5^3 \times 384 = 48,000$, not 128k. Different datasets adjust hidden width and grid size to balance capacity and cost.
> > > * **Empirical evidence** supports meaningful structure: linear PCA on the whole QM9 test molecules (13,388 molecules) shows variance spread across many directions **(52.8% explained by first 32 PCs, 66.7% by 80 PCs)**, indicating the latent space is **not redundant**.
> > > * Nonlinear reconstruction-vs-grid-size ablation confirms spatially informative latent codes. For different L, grid spacing is adjusted so all grids cover the same 3D extent (cube edge = 12 Å), isolating the effect of resolution:
> > >
> > > | $L^3$ grid cells | AE hidden 384 - Success (%) | Mean RMSD (Å) | AE hidden 1024 — Success (%) | Mean RMSD (Å) |
> > > | :-: | :-: | :-: | :-: | :-: |
> > > |     125 (5³)     |     99.2    |     0.054     |             99.5             |     0.031     |
> > > |      64 (4³)     |     81.3    |     0.107     |             87.2             |     0.083     |
> > > |      27 (3³)     |     64.9    |     0.204     |             71.8             |     0.120     |
> > >
> > > **Takeaway:** The latent space contains meaningful structure and is not redundant, addressing efficiency concerns for latent diffusion.
> > >
> > >
> > > ### **Q2: Discrete vs Continuous**
> > >
> > > The latent grid is discrete, but the decoder outputs a **continuous, queryable field**, allowing reconstruction at arbitrary locations. Ablation over L in the above content shows higher resolution significantly improves reconstruction, supporting the design.
> > >
> > >
> > > ### **Q3: Sampling Time / Computational Cost**
> > >
> > > - **Hardware & source:** All times are **measured by us on one RTX 4090** using each method’s official code; **not** copied from other papers.
> > > - **Protocol:** **N = 10** molecules per method, **same GPU**; per-repo mini-batches differ, so we align **N** and hardware. **Reported time = mean over 3 runs.**
> > > - **Checkpoints:** Use **released weights** when available; otherwise **brief training** under the repo’s **default/recommended config** (e.g. VoxMol) **for timing only**.
> > >
> > > ### **Q4: Bond-Level Metrics on GEOM-drugs**
> > >
> > > Reviewer noted VecMol’s bond W1 (~0.104) is worse than baselines. We clarify:
> > >
> > > * **High bond W1 arises from AE reconstruction errors**, not diffusion or latent representation. Increasing AE hidden width improves bond/topology metrics:
> > >
> > > | NF checkpoint | Recon. success (%) | Mean RMSD (Å) | Bond Length W1 |
> > > | :--: | :-: | :-: | :-: |
> > > |   hidden 384  | 79.3  |  0.256  | 0.083 |
> > > |  hidden 1024  | 90.8  |  0.056 | 0.024 |
> > >
> > > * RMSD is computed on successfully reconstructed molecules. Higher AE capacity reduces coordinate errors and bond deviations.
> > > * **Implication:** The metric is reliable; high W1 reflects AE limits rather than latent/diffusion failure.
> > >
> > >
> > > ### **Q5: VecMol_NF vs VecMol**
> > >
> > > * VecMol_NF (reconstruction) and VecMol (generation) serve **different tasks**. Identical bond TV indicates **AE/reconstruction is the main bottleneck**, not diffusion.
> > > * Increasing AE hidden size improves reconstruction and downstream drug-relevant metrics, as shown:
> > >
> > > | Dataset     | Stable Mol (%) | Stable Atom (%) | Validity (%) | Uniqueness (%) |   Valency W₁  |    Atoms TV   |    Bonds TV   | Bond Length W₁ | Bond Angle W₁ | Single Fragment (%) | Median Strain Energy |  Ring Size TV | Atoms per Mol TV |      QED      |      SA     |     logP    |
> > > | :- | :-: | :-: | :-: | :-: | :-: | :-: | :-: | :-: | :-: | :-: | :-: | :-: | :-: | :-: | :-: | :-: |
> > > | hidden 384  |   77.5 ± 0.3   |    98.9 ± 0.1   |  82.5 ± 0.3  |   80.9 ± 0.5   | 0.099 ± 0.004 | 0.033 ± 0.001 | 0.272 ± 0.002 |  0.083 ± 0.02  |  8.44 ± 0.22  | 67.27 ± 0.64 | 58.1 ± 2.4| 0.434 ± 0.065 | 0.250 ± 0.003| 0.639 ± 0.000 | 2.66 ± 0.25 | 2.18 ± 0.16 |
> > > | hidden 1024 | 80.2 ± 0.3 | 99.1 ± 0.1 |  89.2 ± 0.4  | 91.2 ± 0.3| 0.108 ± 0.003 | 0.034 ± 0.000 | 0.102 ± 0.001 |  0.024 ± 0.001 |  4.31 ± 0.20  | 81.82 ± 0.95 |  87.3 ± 3.4 | 0.296 ± 0.037 |   0.214 ± 0.003  | 0.653 ± 0.001 | 2.08 ± 0.19 | 2.30 ± 0.17 |
> > >
> > > * **Key observations:**
> > >
> > >   1. Reconstruction RMSD and bond deviations decrease with higher hidden size.
> > >   2. Drug-relevant metrics improve, showing molecules are **chemically meaningful**.
> > >   3. AE capacity, not diffusion, limits bond/topology accuracy.
> > >
> > > In summary, the latent space is meaningful, bond deviations stem from AE capacity, and the model generates chemically valid, drug-like molecules; we respectfully ask the reviewer to consider this when assessing novelty and significance.

---

### Decision · Program_Chairs · 2026-04-30

**Decision:**

Accept (regular)

**Comment:**

This paper presents an original approach to 3D molecule generation by utilizing continuous vector fields and latent diffusion, offering a promising alternative to discrete graph-based and voxel-based models. Although reviewers initially expressed concerns regarding the latent bottleneck and degraded bond-level metrics on larger datasets like GEOM-Drugs, the authors provided comprehensive rebuttals with additional ablations and scaling experiments that sufficiently addressed these issues. Given the clear consensus among reviewers on the work's novelty, technical soundness, and the strength of the rebuttal, I recommend to accept this paper.